# Epigenetic modifications and metabolic gene mutations drive resistance evolution in response to stimulatory antibiotics

Hui Lin[1,2], Donglin Wang[1], Qiaojuan Wang[1,2], Jie Mao[1], Lutong Yang[1,2], Yaohui Bai [ID][1✉] & Jiuhui Qu[1]

## Abstract

**The antibiotic resistance crisis, fueled by misuse and bacterial evolution, is a major global health threat. Traditional perspectives tie resistance to drug target mechanisms, viewing antibiotics as mere growth inhibitors. New insights revealed that low-dose antibiotics may also serve as signals, unexpectedly promoting bacterial growth. Yet, the development of resistance under these conditions remains unknown. Our study investigated resistance evolution under stimulatory antibiotics and uncovered new genetic mechanisms of resistance linked to metabolic remodeling. We documented a shift from a fast, reversible mechanism driven by methylation in central metabolic pathways to a slower, stable mechanism involving mutations in key metabolic genes. Both mechanisms contribute to a metabolic profile transition from glycolysis to rapid gluconeogenesis. In addition, our findings demonstrated that rising environmental temperatures associated with metabolic evolution accelerated this process, increasing the prevalence of metabolic gene mutations, albeit with a trade-off in interspecific fitness. These findings expand beyond the conventional understanding of resistance mechanisms, proposing a broader metabolic mechanism within the selective window of stimulatory sub-MIC antibiotics, particularly in the context of climate change.**

**Keywords** Evolutionary Dynamics; Gene Mutations; Metabolic Remodeling; Methylation Modifications; Stimulatory Antibiotic Resistance
**Subject Categories** Chromatin, Transcription & Genomics; Metabolism

## Introduction

The growing challenge of antibiotic resistance significantly impacts infectious disease management, posing a considerable threat to human health as well as the agricultural and veterinary sectors. A deeper understanding of the mechanisms that facilitate the emergence and rapid spread of antibiotic resistance is essential to efficiently combat these threats. Traditionally, it was posited that exposure to antibiotic concentrations exceeding the minimum inhibitory concentration (MIC)—lethal doses—was the primary catalyst for the development of antibiotic resistance (Drlica, 2003; Gullberg et al, 2011). However, antibiotics are often present at sub-minimal inhibitory concentrations (sub-MIC) in various natural environments such as sewage water, sludge, rivers, lakes, and even drinking water (Baquero et al, 2008; Fram and Belitz, 2011; Jiang et al, 2013; Khan et al, 2013), as well as in patients and livestock during antibiotic therapy (Baquero et al, 1998; Baquero and Negri, 1997; Rice, 2013). At these sub-MIC levels, antibiotics do not kill the susceptible strains but rather allow cells to continue growing, which can also promote the evolution of resistance or tolerance, becoming a new crucial aspect of the current antibiotic resistance crisis (Andersson and Hughes, 2014). These sub-MIC concentrations have been reported to enhance resistance development by increasing resistant bacterial populations (Gullberg et al, 2011), facilitating the dissemination of resistance genes (Laureti et al, 2013), and accelerating mutation rates at drug target sites (Andersson and Hughes, 2014; Baharoglu and Mazel, 2011), leading to robust phenotypic resistance (Andersson and Hughes, 2014; Laureti et al, 2013).

Despite this, much of the existing research on resistance development and mechanisms in bacterial populations has focused on scenarios where sub-MIC antibiotics act solely as growth inhibitors. Recent insights challenge this perspective, revealing that at sub-MIC levels, antibiotics can act as signaling molecules, beneficially affecting bacterial behaviors such as growth, motility, biofilm formation, and secondary metabolite induction (Agathokleous et al, 2022; Iavicoli et al, 2021; Tang et al, 2022; Xu et al, 2023). For instance, low concentrations of ribosome-binding antibiotics are known to enhance the longevity and growth efficiency of *E. coli* populations (Wood et al, 2023); piperacillin can induce secondary metabolite production in *Burkholderia thailandensis* (Li et al, 2021) and neomycin and erythromycin have been shown to boost bacterial bioluminescence in *Vibrio fischeri* via the LuxR quorum sensing (QS) system (Sun et al, 2019; Yao et al, 2019). Whether these stimulatory effects of sub-MIC antibiotics contribute to resistance development and the mechanisms behind these contributions remain unknown.

Often linked with elevated metabolic states, the stimulatory effects of sub-MIC antibiotics create a unique selective environment that intensifies metabolism-specific selective pressures. Various studies showed that the metabolic state of a bacterial cell

[1]Research Center for Eco-Environmental Sciences, Chinese Academy of Sciences, 100085 Beijing, China. [2]University of Chinese Academy of Sciences, 100049 Beijing, China.
✉E-mail: yhbai@rcees.ac.cn

influences its susceptibility to antibiotics, with certain states enhancing or reducing vulnerability (Gutierrez et al, 2017; Peng et al, 2015b; Zhao et al, 2021). Recent research provides insights into the metabolic states of persisters within biofilms, which offer a protective niche against antibiotics and other adverse conditions (Davenport et al, 2014; Donlan and Costerton, 2002). These cells often undergo metabolic downshifts, exacerbated by impaired nutrient penetration and consumption by peripheral cells, leading to reduced nutrient availability and possibly benefiting long-term survival through impaired metabolism. In *M. tuberculosis*, increased antibiotic tolerance and persistence have also been correlated with growth arrest and changes in carbon flux (Baek et al, 2011). Such complexity of metabolic responses suggests a range of potential evolutionary outcomes beyond those associated with canonical drug targets, highlighting the potential role of underappreciated noncanonical mechanisms, particularly those involved in central carbon and energy metabolism, in fostering antibiotic resistance (Wood et al, 2023). However, mutations conferring resistance are rarely found in metabolic genes, and the role of metabolic regulation as a resistance mechanism has been seldom addressed (Lopatkin et al, 2021). Therefore, understanding how stimulatory sub-MIC antibiotics shape bacterial resistance evolution and drive the emergence of novel genetic resistance mechanisms, especially those linked to metabolic strategies, is crucial.

Using concentration-response curves (CRCs), we defined the zero effective concentration point (ZEP)—the turning point between inhibitory and stimulatory effects—as a marker of resistance for *Comamonas testosteroni* when exposed to stimulatory levels of sulfamethoxazole (SMX). Our comprehensive analysis spanned resistance phenotypes, genetic and epigenetic modifications, and mutant growth fitness (Appendix Fig. S1), unveiling a novel genetic mechanism of antibiotic resistance: the remodeling of core metabolic pathways via both epigenetic and genetic modifications. Under a dynamic SMX evolution protocol, resistance displayed oscillatory dynamics, transitioning from a fast and transient mechanism (FTM) driven by methylation in central metabolic pathways to a slow and stable mechanism (SSM) characterized by mutations in core metabolic genes. This evolution marked a shift from acquired to intrinsic resistance, consistently leading to a gluconeogenic profile favoring rapid growth. Of note, increasing temperatures associated with metabolic evolution expedited the evolutionary process, enhancing metabolic gene mutation rates and the prevalence of mutations, albeit at the cost of interspecific fitness. These insights are vital for devising new strategies to combat antibiotic resistance, emphasizing the importance of considering broader ecological and evolutionary perspectives, particularly in the context of climate change.

## Results

### Evolutionary trajectories of phenotypic resistance under static and dynamic conditions

In previous research, we established that SMX differentially affects the cell growth of *C. testosteroni* across a range of concentrations, with stimulatory effects at low doses (<250 μg·L$^{-1}$) and inhibitory effects at high doses (>250 μg·L$^{-1}$), referred as hormetic effects (Lin

et al, 2023). Therefore, this bacterium serves as an ideal model organism for studying the evolution of phenotypic resistance under stimulatory stress. We subjected the *C. testosteroni* CNB-2 strain to evolution experiments with and without SMX exposure, utilizing both static and dynamic protocols. In the dynamic protocol, the concentration of SMX was incrementally increased by 100 μg·L$^{-1}$ daily, starting from an initial dose of 100 μg·L$^{-1}$, which stimulated 50% of cell growth relative to SMX-free conditions, reaching 5.5 mg·L$^{-1}$ by day 55 (Fig. 1A). The static protocol maintained a constant SMX concentration of 100 μg·L$^{-1}$ throughout the experiment (Fig. 1A).

To analyze resistance under the hormetic effects of antibiotics, we adopted the ZEP as a novel marker for bacterial resistance (Fig. EV1A). This metric builds upon the No Observable Adverse Effect Level (NOAEL), marking the transition from stimulatory to neutral impacts of antibiotics on bacterial growth (Dorato and Engelhardt, 2005). The ZEP specifically identifies a threshold where SMX begins to modulate bacterial physiology—impacting metabolic activity, gene expression, and stress response mechanisms—without suppressing growth (Agathokleous et al, 2021; Calabrese, 1996; Kendig et al, 2010). We determined the ZEP by analyzing CRCs using the Hill inverse dichotomy iteration model. We speculated that resistance would be continually selected for over repeated SMX treatments, increasing the ZEP of the evolved population. As expected, resistance in SMX-treated cultures increased significantly early on. By day 5, there was a 3.92-fold rise in the ZEP for statically evolved strains (Wilcoxon $P < 0.01$, Fig. 1B) and a 4.64-fold increase for dynamically evolved strains compared to the control evolved strain without SMX treatment (Wilcoxon $P < 0.01$, Fig. 1C). Resistance under static evolution steadily increased, reaching a peak ZEP of $25.3 \pm 1.5$ mg·L$^{-1}$, marking an over 100-fold increase (Figs. 1B and EV1B). Conversely, dynamic exposure led to a unique resistance pattern characterized by oscillations in ZEP, with peaks of resistance (days 15 and 25) followed by sharp declines, before stabilizing at a high level of resistance (Figs. 1C and EV1C). The oscillatory resistance patterns observed in dynamically evolving strains demonstrated the emergence of acquired resistance, followed by its rapid reversal to susceptibility in new environments. This pattern suggested a potential transition from unstable, diverse epigenetic mechanisms to stable, mutation-induced intrinsic resistance. In addition, our analysis of changes in the MIC during the evolutionary process revealed a pattern of increase similar to that observed with the ZEP, further substantiating the progression of resistance (Fig. EV2).

We also measured per capita growth rates of both ancestral and evolved strains across SMX concentrations from 0 to 100 mg·L$^{-1}$ to rule out the impact of reduced growth rates on SMX susceptibility. Results demonstrated that these strains consistently exhibited higher growth rates below the ZEP compared to SMX-free conditions (Appendix Fig. S2 and Appendix Table S1). Additionally, strains evolved under SMX conditions consistently outperformed those evolved without SMX across all tested concentrations (Appendix Table S1). A significant positive correlation between per capita growth rate and biomass across all concentrations further confirmed that the observed enhanced resistance was not due to reduced growth rates (Fig. EV3A), but rather reflected an inherent evolutionary advantage in strains evolved under SMX exposure. This advantage was further

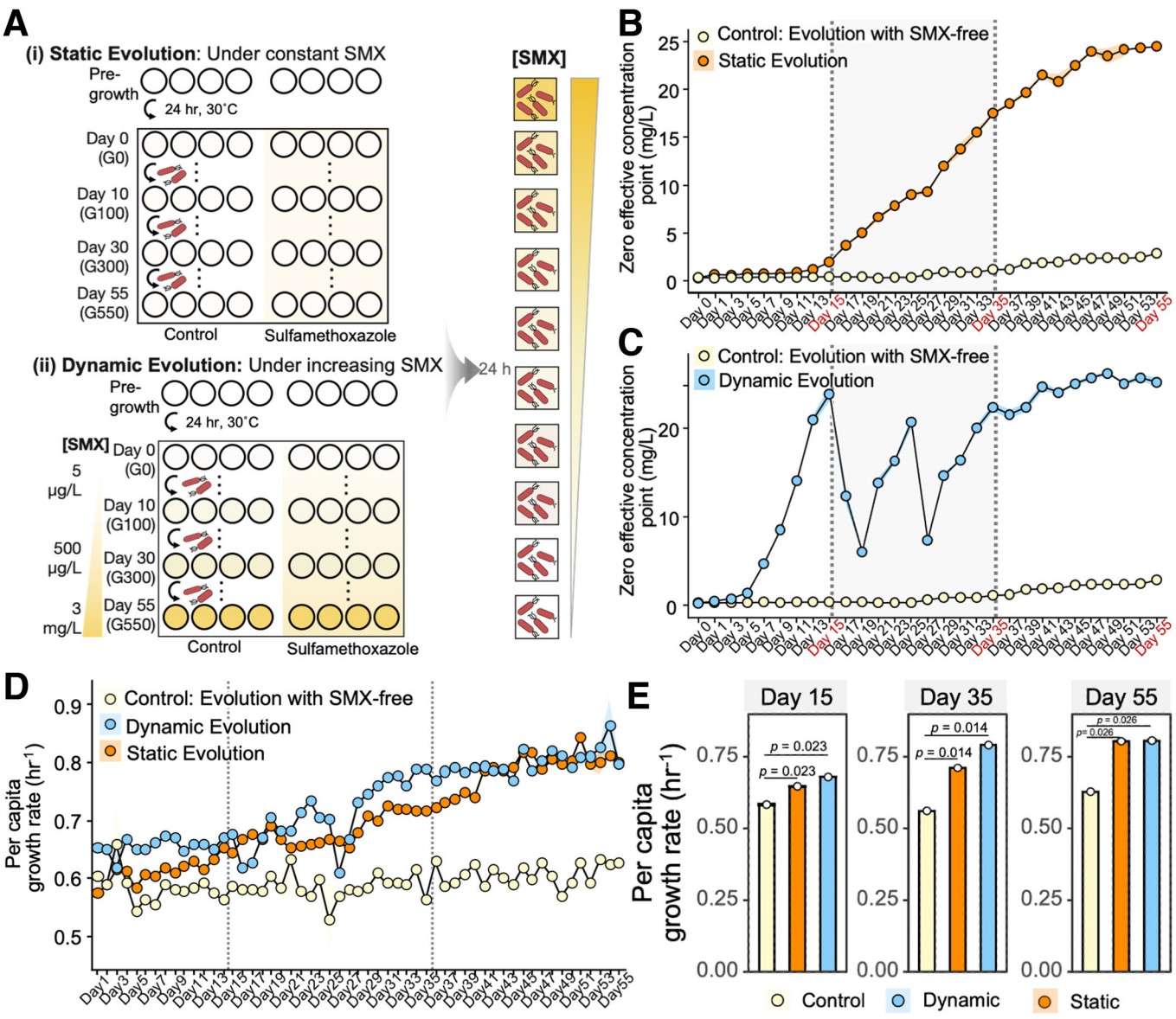

**Figure 1. Evolutionary dynamics and antibiotic resistance development in *Comamonas testosteroni* under sulfamethoxazole (SMX) exposure.**

(A) Evolution schematic. Four replicate populations were inoculated and grown at 30 °C in NB Medium, with $OD_{600}$ measured every 24 h and MICs every 48 h. In the dynamic protocol, SMX concentrations increased incrementally by 100 µg·L⁻¹ daily, starting from an initial dose of 100 µg·L⁻¹, reaching 6 mg·L⁻¹ by day 55, while the static protocol maintained a consistent SMX concentration (100 µg·L⁻¹). (B) Incremental increase in zero effective concentration point (ZEP), indicating resistance development in statically evolved strains over time compared to control. (C) Resistance patterns for dynamically evolved strains, characterized by oscillations in ZEP. (D, E) Enhanced growth rate in SMX-evolved cultures relative to control evolved groups at days 15 (G150), 35 (G350), and 55 (G550). Data were expressed as mean ± cumulative standard deviation of six biological replicates. Significance levels were assessed using Wilcoxon test. The per capita growth rate referred to the population growth rate normalized by the initial population size. Source data are available online for this figure.

supported by our data, which demonstrated that biomass production and intrinsic growth rates of evolved strains under both static and dynamic protocols consistently increased, surpassing those of control evolved strains despite stable nutrient conditions (Fig. 1D,E; Appendix Figs. S3 and S4 and Appendix Table S1). A significant correlation between per capita growth rate and end biomass across all evolved strains (G0–G550) was also observed (Fig. EV3B). In contrast, control strains evolved under SMX-free conditions did not show increases in resistance levels or growth rates, suggesting that standard growth conditions, such as liquid medium and 24 h

passaging, do not naturally facilitate the evolution of antibiotic resistance.

## Identifying epigenetic modifications associated with antibiotic-acquired resistance in the oscillation phase

Unlike epigenetic mechanisms in eukaryotes, bacteria do not possess histones but contain a variety of enzymes capable of applying epigenetic changes, specifically through sequence-specific methylation of DNA bases (Adhikari and Curtis, 2016; Casadesús

and Low, 2013; Lin et al, 2023). We therefore explored the influence of DNA methylation on SMX resistance evolution by utilizing SMRT sequencing to analyze genome-wide methylation patterns. Our analysis focused on evolved cultures isolated from days 15 (G150) and 25 (G250) under both static and dynamic SMX exposure scenarios, using the ancestral strain that evolved under SMX-free conditions as the control. Results revealed that statically evolved isolates exhibited methylation patterns similar to the ancestral controls (Appendix Figs. S5 and S6), whereas dynamically evolved isolates demonstrated notable changes, specifically decreased methylation gene counts associated with the "Cell wall/ membrane/envelope biogenesis" pathway and increased counts associated with the "Carbohydrate transport and metabolism" pathway (Fig. 2A; Appendix Fig. S6). These findings suggested that modifications in central metabolism genes play a role in the development of acquired resistance in dynamically evolved *C. testosteroni* isolates. Moreover, within the "Cell wall/membrane/ envelope biogenesis" pathway, specific genes that ceased to be methylated were linked to various categories, including genetic antibiotic targets, cell membrane permeability related to antibiotic uptake, and multi-drug resistance pumps (Appendix Tables S2 and S3). These pathways are traditionally recognized in the context of antibiotic resistance mechanisms and have been identified as potential therapeutic targets for combating SMX resistance.

Generally, DNA methylation is associated with gene silencing and plays a crucial role in regulating mRNA transcription. To investigate the relationship between methylation and gene expression levels, and to further delineate the role of central metabolic gene modifications, RNA-seq was performed on the G150 and G250 dynamically evolved isolates. We identified the intersection of differentially methylated genes (DMGs) and differentially expressed genes (DEGs) and conducted genomic enrichment analysis (GSEA) to explore their functional implications (Fig. 2B, top). Results showed significant metabolic alterations in the "TCA cycle" and "Pentose phosphate pathway" with substantial downregulation, and upregulation in the "Glycolysis/Gluconeogenesis" pathway for both G150 and G250 isolates (Fig. 2B, bottom, (false discovery rate [FDR] *q*-value < 0.01). Specifically, dynamically evolved isolates exhibited significant downregulation in key TCA cycle enzymes such as aldehyde dehydrogenase (*ALDH*), citrate synthase (*GltA*), and citrate lyase (*AcnB*), as evidenced by changes in expression levels (Fig. 2C, |log₂FC| > 1.5, [FDR] *q*-value < 0.01). Furthermore, in the gluconeogenesis pathway, we observed a significant 2.2-fold reduction in pyruvate kinase (*PK*) levels and a twofold increase in phosphoenolpyruvate dehydrogenase (*PEPs*) levels (Fig. 2C, FDR *q*-value < 0.01). The changes at the *PK/PEPs* node in G150 and G250 isolates indicated a specific shift in carbon flux towards gluconeogenesis. This regulation of gluconeogenic carbon flux, driven by changes in DNA methylation patterns and altered expression of metabolic genes, could tip the metabolic balance toward increased biomass production while simultaneously reducing resource allocation to less essential pathways.

Furthermore, we used transmission electron microscopy (TEM) to explore the phenotypic changes in the evolved isolates. Granular bodies, presumed to be poly-3-hydroxybutyrate (PHB), were visible in dynamically evolved G250 resistance isolates, but absent in the control and statically evolved isolates (Fig. 2D). Cross-sectional TEM images revealed the presence of these PHB granules, oval and round in shape, within the cell lumen and adjacent to the cytoplasmic membrane, encapsulated by a monolayer of proteins and phospholipids critical for the synthesis and degradation of PHB and formation of granules (Pötter and Steinbüchel, 2005). Previous studies have indicated that PHB can serve as a carbon and energy source for cells, influencing the intracellular metabolic flow and the redox state and enhancing cellular stress resistance (Wang et al, 2009). Consequently, PHB accumulation, alongside increased biomass production, may act as a mechanism for producing amino acids and essential metabolic compounds (Gu et al, 2013; Kang et al, 2010; Liu et al, 2007), thereby inducing resistance against adverse environmental conditions.

## Identifying genetic changes associated with antibiotic intrinsic resistance during evolution

Next, we investigated the slow and steady intrinsic resistance mechanisms emerging late in evolution. Whole-population sequencing traced the mutational dynamics within evolved isolates at days 15 (G150), 35 (G250), and 55 (G550) under static and dynamic SMX exposure, using the SMX-free control evolved strain as a reference. Valid mutant genes were identified using standardized variant calling, focusing on non-synonymous SNPs and InDels that alter the amino acid sequence and requiring that the frequency of a mutant gene exceed 10% in any replicate. Analysis of SNPs in both statically and dynamically evolved isolates revealed that most of the high-frequency mutations (e.g., present in >90%) were noncanonical and related to central metabolic processes. At the evolutionary endpoint (G550), all evolved isolates, both static and dynamically evolved, exhibited mutations in genes including *ALDH*, *GltA*, *AcnB*, *Bug*, *Pck*, *NadB*, and *TctA* (Fig. 3A), pivotal for encoding metabolic enzymes responsible for the oxidation of ACoA and citrate in the TCA cycle or associated with their transport (Fig. 3B). Indeed, initiation of all catabolic pathways is dependent on the transport of substrates from the external environment to the cell interior (Rosa et al, 2018; Wang et al, 2023), a process facilitated by specific membrane transporters. Among them, *TctC* is crucial for low-μM affinity citrate uptake, with mutations in this gene found to significantly disrupt citrate transportation (Rosa et al, 2018). Moreover, mutations were discovered in other essential metabolic-related genes such as *Pck*, encoding a key cataplerotic enzyme in gluconeogenesis; *ALDH*, involved in acetate processing; and *NadB*, responsible for converting L-aspartate into quinoline, an essential precursor for NAD synthesis. Given that NAD is crucial for energy metabolism, cellular defense, and biosynthesis, with the NAD⁺/NADH redox pair serving as an important regulator of these processes (Prunier et al, 2007), the mutations observed highlight significant shifts in cellular metabolic functions. Notably, the emergence of metabolic mutations occurred later in static evolution compared to dynamic scenarios, where most high-frequency mutations appeared by G150. Under static conditions, comparable high-frequency mutations were not observed until G550. In addition, the mutated genes aligned with those showing altered DNA methylation in the dynamically evolving G250 acquired resistance isolates. This pattern supported our hypothesis: under rapidly increasing antibiotic pressure, bacteria initially exhibit acquired resistance—a transitory state regulated by DNA methylation of core metabolic genes—which is eventually stabilized by mutations in these genes.

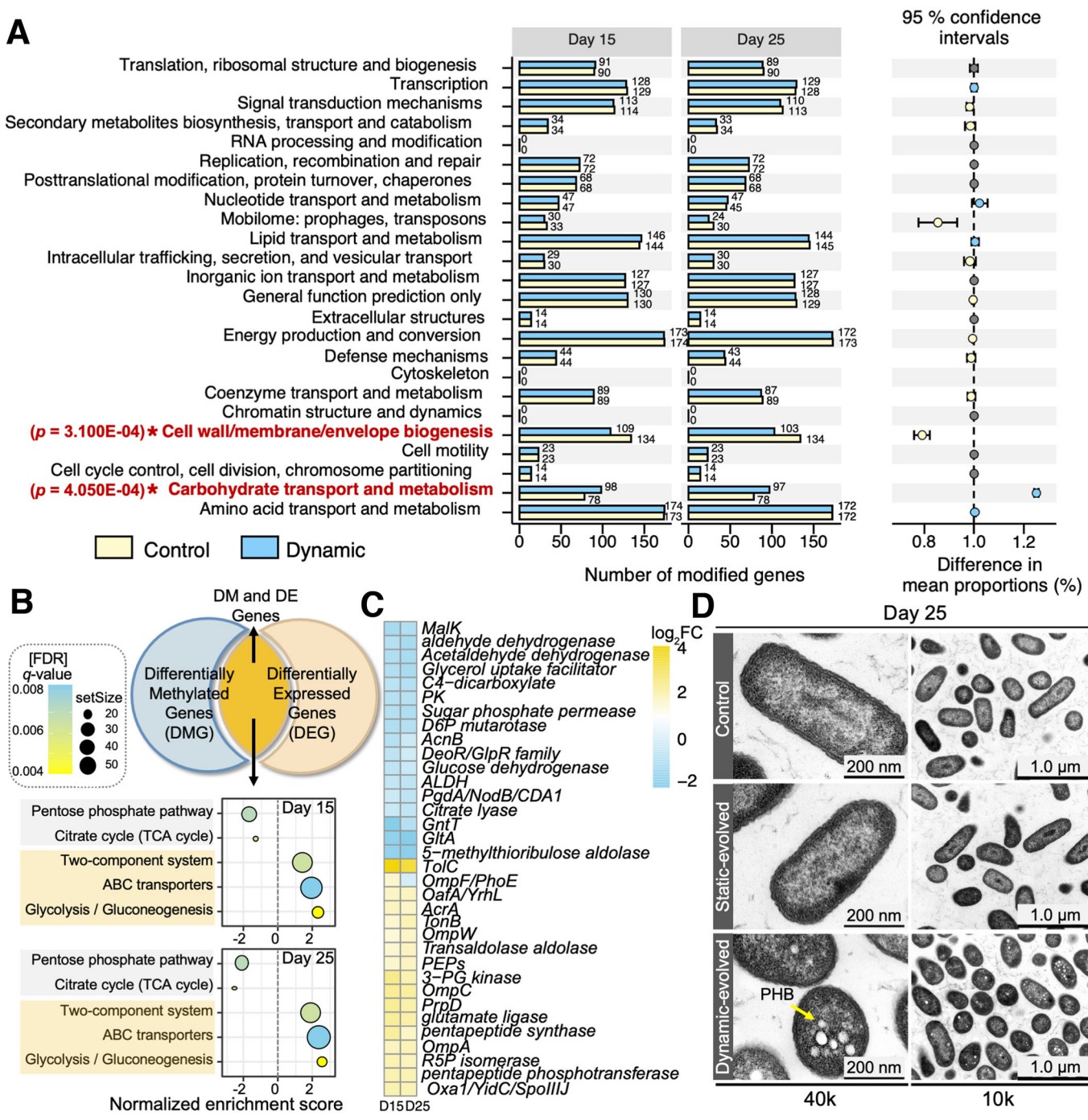

**Figure 2. Analysis of DNA methylation impact on acquired resistance evolution in *Comamonas testosteroni* during oscillation phase.**

(A) Comparative KEGG pathway analysis of DNA methylation patterns in resistant isolates from days 15 (G150) and 25 (G250) under dynamic conditions, using control evolved strains as a reference ($n = 3$). Statistical significance was determined using a permutation test (1000 permutations), and False Discovery Rate (FDR) was applied to adjust $P$ values for multiple comparisons. (B) Identification of enriched KEGG pathways in DMGs and DEGs between dynamically evolved isolates G150 and G250 and control evolved strains ($n = 3$). (C) Heatmap of changes in gene expression ($\log_2$FC) and identification of significant changes ($|\log_2$FC$| > 1$, [FDR] $q$-value $< 0.01$) in DM and DE genes ($n = 3$). (D) TEM (scale bar: 1 μm) and zoomed-in TEM (scale bar: 200 nm) images of dynamically evolving and control isolates on day 25. Yellow arrows represented poly-3-hydroxybutyrate (PHB). Source data are available online for this figure.

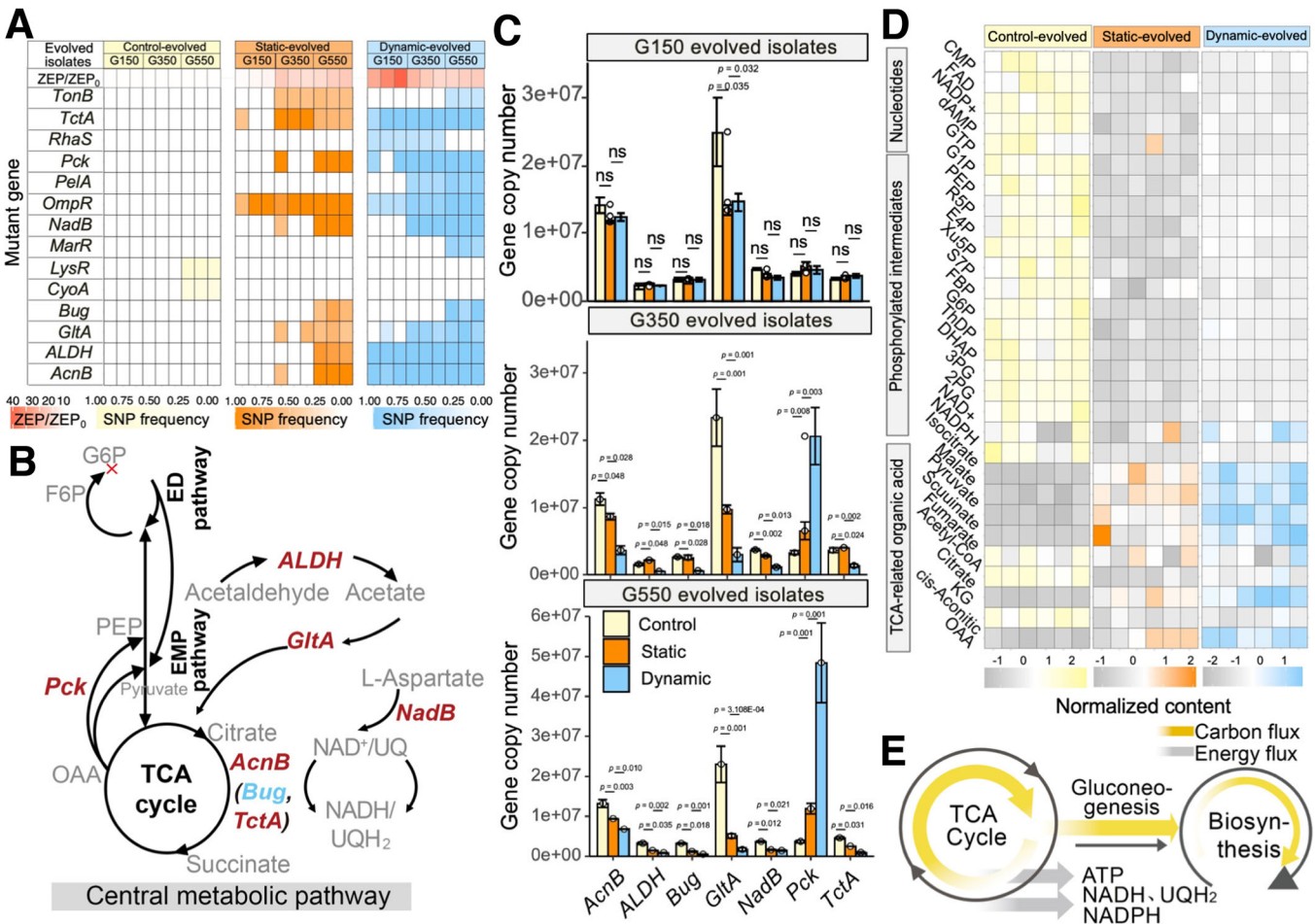

**Figure 3.    Comprehensive analysis of sulfamethoxazole (SMX)-induced genetic and metabolic changes.**

(**A**) Summary of gene mutations, frequencies of mutant genes in evolved isolates from days 15 (G150), 35 (G350), and 55 (G550). ZEP/ZEP$_0$ represents the relative resistance level of three whole-genome sequenced isolates compared to the control evolutionary strain. (**B**) Impact of mutations on central metabolic pathways, indicating significant metabolic reprogramming. Red: SNP; Blue: InDel. (**C**) Changes in gene expression assessed by qRT-PCR, revealing an increase in cataplerotic enzymes such as *Pck* and a decrease in other metabolic enzymes. Data were expressed as mean ± cumulative standard deviation of three biological replicates and two technical replicates. Significance levels were assessed using Wilcoxon test. (**D** Targeted LC-MS/MS metabolomics analysis of central carbon metabolites in evolved G550 isolates, with significance determined using one-way ANOVA followed by Tukey HSD post hoc test ($n = 6$). (**E**) Diagram of altered gluconeogenic carbon fluxes and associated energy shifts, showing a gluconeogenic growth phenotype in evolved isolates. Source data are available online for this figure.

Subsequently, using quantitative reverse transcriptase polymerase chain reaction (qRT-PCR), we assessed the impact of these mutations in metabolic genes on their expression levels. Results indicated a significant increase in the relative abundance of cataplerotic enzymes (PEP carboxykinase, *Pck*) at the crucial metabolic intersection between PEP and oxaloacetate (OAA). The expression of *Pck* gene showed a significant 3.71-fold and 3.57-fold increase in statically evolved G350 and G550, and a significant 6.13-fold and 12.93-fold increase in dynamically evolved G350 and G550, respectively (Fig. 3C, Wilcoxon $P < 0.01$). Besides, the expression levels of other key mutant metabolic enzymes were markedly reduced, showing a 1.36–4.76-fold decrease in statically evolved G550 and a 2.0–12.34-fold decrease in dynamically evolved G550 (Fig. 3C, Wilcoxon $P < 0.01$). Therefore, genetic mutants may influence metabolism-related gene expression, leading to an increased presence of the cataplerotic enzyme at the key metabolic transition between OAA and PEP, thereby modulating carbon flow through the TCA cycle and contributing to resistance.

To further identify shifts in the allocation of carbon fluxes in metabolic pathways, targeted metabolomics utilizing UPLC-MS/MS was employed to profile the metabolite pools of G550 isolates. Among the 29 intracellular metabolites profiled, there was a consistent reduction in abundances of all nucleotides and phosphorylated intermediates—reaching up to 100%—across biological replicates of both statically and dynamically evolved isolates relative to ancestral evolved isolates (Fig. 3D; Appendix Fig. S6). Lower concentrations of phosphorylated intermediates of evolved isolates (up to 4.75-fold in 2PG, Tukey HSD post hoc test $P < 0.01$) indicated diminished carbon accumulation in the ED and PP pathways among isolates with higher resistance. In contrast, the cumulative pools of TCA cycle-related organic acids were 60% higher in both statically and dynamically evolved isolates (Tukey HSD post hoc test $P < 0.01$). Overall, the relative metabolite compositions in cells under the two different evolutionary conditions revealed a shift in carbon accumulation in selective

pathways, enhancing carbon availability in the TCA cycle in resistance-evolved cells, as opposed to increased carbon in the upper EMP pathway and limited carbon availability in the TCA cycle in ancestral evolved cells (Fig. 3D). Notably, nucleoside triphosphates were significantly depleted in resistance-evolved cells compared to ancestral cells, yet the lowest energy charge in G550 cells (0.65 ± 0.03) remained above the minimum required for cell viability (Appendix Fig. S7) (Chapman et al, 1971; Kayser et al, 2005).

To address potential confounding factors of overflow metabolism, we quantified typical byproducts such as acetate, ethanol, and lactate in the G550 isolates. These compounds were undetectable in both static and dynamic conditions, suggesting no overflow metabolism, and all isolates showed increased growth rates and biomass yield (Appendix Table S4). Physiological assessments further confirmed that oxygen consumption rates did not decrease in these isolates (Appendix Fig. S8a). In addition, unlike the control evolved isolates, neither static nor dynamic isolates exhibited elevated $NAD^+$/NADH ratios, which typically indicate hyperactive TCA cycle activity under oxidative stress (Appendix Fig. S8b). Taken together, resistance-evolved cells undergo a critical metabolic shift from glycolysis to gluconeogenesis. This remodeling, along with the increased biosynthetic demands of gluconeogenesis, consumes both carbon and ATP, limiting the carbon available for metabolite secretion or futile cycling through the EMP and PP pathways (Fig. 3E; Appendix Table S4). Consequently, mutations in metabolic genes allowed the resistance-evolved isolates to optimize their metabolic patterns, thereby enhancing their intrinsic resistance.

## Evolving antibiotic resistance using a metabolic-dependent approach

The discovery of metabolic alterations underscores their crucial, yet previously underestimated, role in the development of resistance to antibiotic interventions. Classical evolution experiments with continuous antibiotic exposure typically select for growth-dependent traits, often at the expense of metabolic-specific adaptations due to the absence of targeted selective pressures. To prioritize metabolic adaptations and further teased out resistance arising from metabolic variations, we adopted an alternative metabolic evolution protocol (Lopatkin et al, 2021), subjecting bacteria to constant SMX exposure while progressively increasing metabolic activities (Fig. 4A). During antibiotic treatment, metabolic activity was modulated through temperature adjustments, a well-documented factor influencing basal metabolic rates (Price and Sowers, 2004), which was consistently maintained throughout each SMX evolution cycle. Temperature was increased daily in increments of 1 °C (18–37 °C from days 0 to 55). Otherwise, the protocol remained identical to that shown in Fig. 1A. The optimal temperature range of 18 °C to 37 °C, based on the wild-type *C. testosteroni* growth rate response curve (Appendix Fig. S9), aligns with the organism's ecological tolerance and typical environmental conditions, enabling metabolic optimization to aid in identifying specific mutants.

In line with the static evolution experiment, the metabolic evolution experiments demonstrated a monotonical increase in ZEP, growth rate, and biomass production compared to the ancestral strain, surpassing the increases observed under traditional

static evolution (Fig. 4B,C; Appendix Fig. S3). Sequencing was performed on the G150, G350, and G550 isolates undergoing metabolic evolution and compared to the ancestral strain that evolved under SMX-free conditions. Analysis using the established pipeline revealed that metabolic evolution isolates exhibited high frequencies of specific mutations exclusively in core metabolic genes, the same genes identified in our classical evolutionary experiments (Fig. 4D). providing additional evidence of significant shifts in metabolic pathways. Since many of these mutations were detected at high frequencies (90–100%), the metabolically evolved isolates were isolated and sequenced to confirm that they carried mutations in *TctA*, *Pck*, *NadB*, *GltA*, *Bug*, *Lpd*, *ALDH*, and *AcnB*. Furthermore, the qRT-PCR results, which measured the expression levels of these metabolic genes, were consistent with observations from classical evolutionary isolates (Appendix Fig. S10). This consistency provides additional evidence of significant shifts in metabolic pathways, supporting the link between genetic mutations and functional metabolic changes.

To determine whether metabolic mutations directly confer resistance or are secondary effects of other mechanisms, we reversed mutations in specific metabolic genes (*TctA*, *Pck*, *NadB*, *GltA*, *ALDH*, and *AcnB*) found in both classically evolved and metabolically evolved isolates. These mutations were reversed in metabolically evolved isolates, where SNPs in these genes had been identified. We employed the suicide plasmid pCVD442GS, which carries the wild-type versions of these genes, to revert the mutations in the evolved isolates (Appendix Table S5). The plasmid facilitates the expression and integration of the wild-type genes into the genome of the recipient strain, *E. coli* β2155 (Fig. 4E), followed by conjugation with the metabolically evolved recipient strain. Subsequent comparisons of the ZEP between the reverted mutants and their metabolically evolved counterparts revealed significant reductions for all reverted mutants. Notably, there was a 16.98-fold decrease for the reverted mutants of the key cataplerotic enzyme in gluconeogenesis, $Pck^M$, and a 14.85-fold decrease for the reverted mutants of $AcnB^M$ (Fig. 4F). Per capita growth rate measurements further supported these reults, with reverted mutants of $Pck^M$ and $AcnB^M$ displaying decreases of 1.52-fold and 1.49-fold, respectively (Fig. 4G). These results confirmed that the identified metabolic mutations indeed directly contribute to resistance. Further investigations are required to determine whether these mutations modify catalytic activity or alter the structure and function of proteins.

## Gluconeogenic carbon substrates promote resistance compared to glycolytic substrates

Recognizing that metabolic shifts can influence SMX resistance, we investigated whether wild-type strains could alter their metabolic strategies to enhance resistance, independent of genetic mutations. We assessed the resistance of both wild-type *C. testosteroni* and its metabolically evolved variants when cultured on succinate and gluconate, which represent gluconeogenic and glycolytic metabolic pathways, respectively (Wilkes Rebecca et al, 2022). Succinate metabolism directly engages gluconeogenesis from the TCA cycle to support cellular functions, facilitated by a C4-dicarboxylate transporter, with NADPH production tied to TCA cycle post-isocitrate dehydrogenase activity (Beauprez et al, 2011; Rhie et al, 2018). In contrast, gluconate, undergoes conversion into

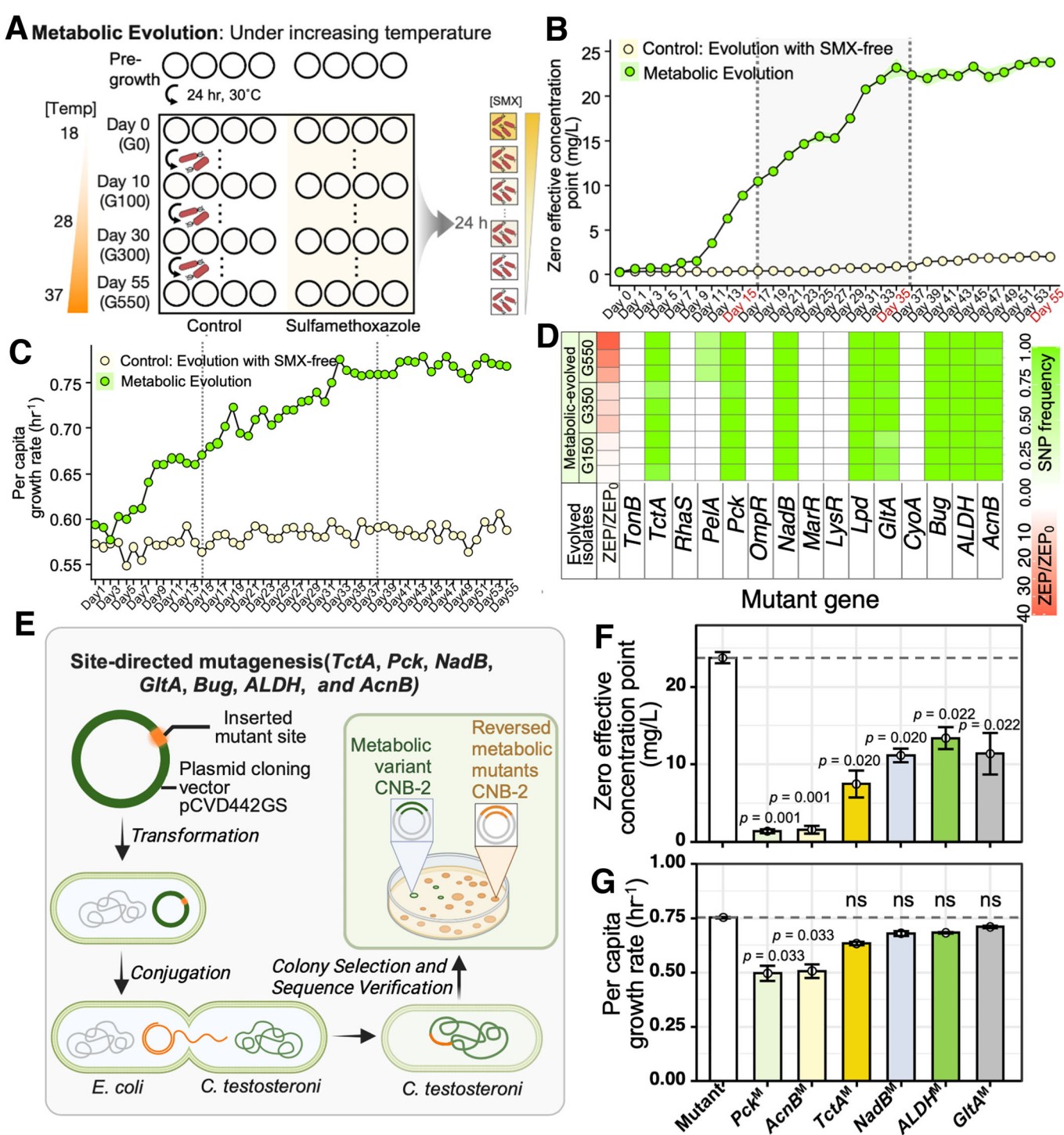

**Figure 4. Metabolic evolution directly enhances antibiotic resistance development.**

(A) Schematic of metabolic evolution experiment: continuous sulfamethoxazole exposure with progressive metabolic activity increases, via daily temperature increments from 18 °C to 37 °C, over 55 days under conditions identical to those in Fig. 1a. (B, C) Data depicted a steady increase in zero effective concentration point (ZEP) ($n = 3$) and per capita growth rate ($n = 6$) in metabolically evolved strains compared to control evolved strains, outperforming standard static evolution. The per capita growth rate referred to the population growth rate normalized by the initial population size. (D) Mutation analysis showed increased frequencies of mutant genes in core metabolic genes within evolved isolates from days 15 (G150), 35 (G350), and 55 (G550) ($n = 3$). ZEP/ZEP₀ represents the relative resistance level of three whole-genome sequenced isolates compared to the control evolutionary strain. (E) Schematic of site-directed mutagenesis used to reverse mutations. Subfigures (F) and (G) displayed the ZEP and growth rates, respectively, of reversed metabolic gene mutants with SNPs ($Pck^M$, $AcnB^M$, $TctA^M$, $NadB^M$, $ALDH^M$, and $GltA^M$) ($n = 3$ for ZEP; $n = 6$ for growth rate). ZEP data were expressed as the mean ± standard deviation (SD) of three biological replicates, while growth rate data were expressed as the mean ± SD of six biological replicates. Significance levels were assessed using Wilcoxon test. Source data are available online for this figure.

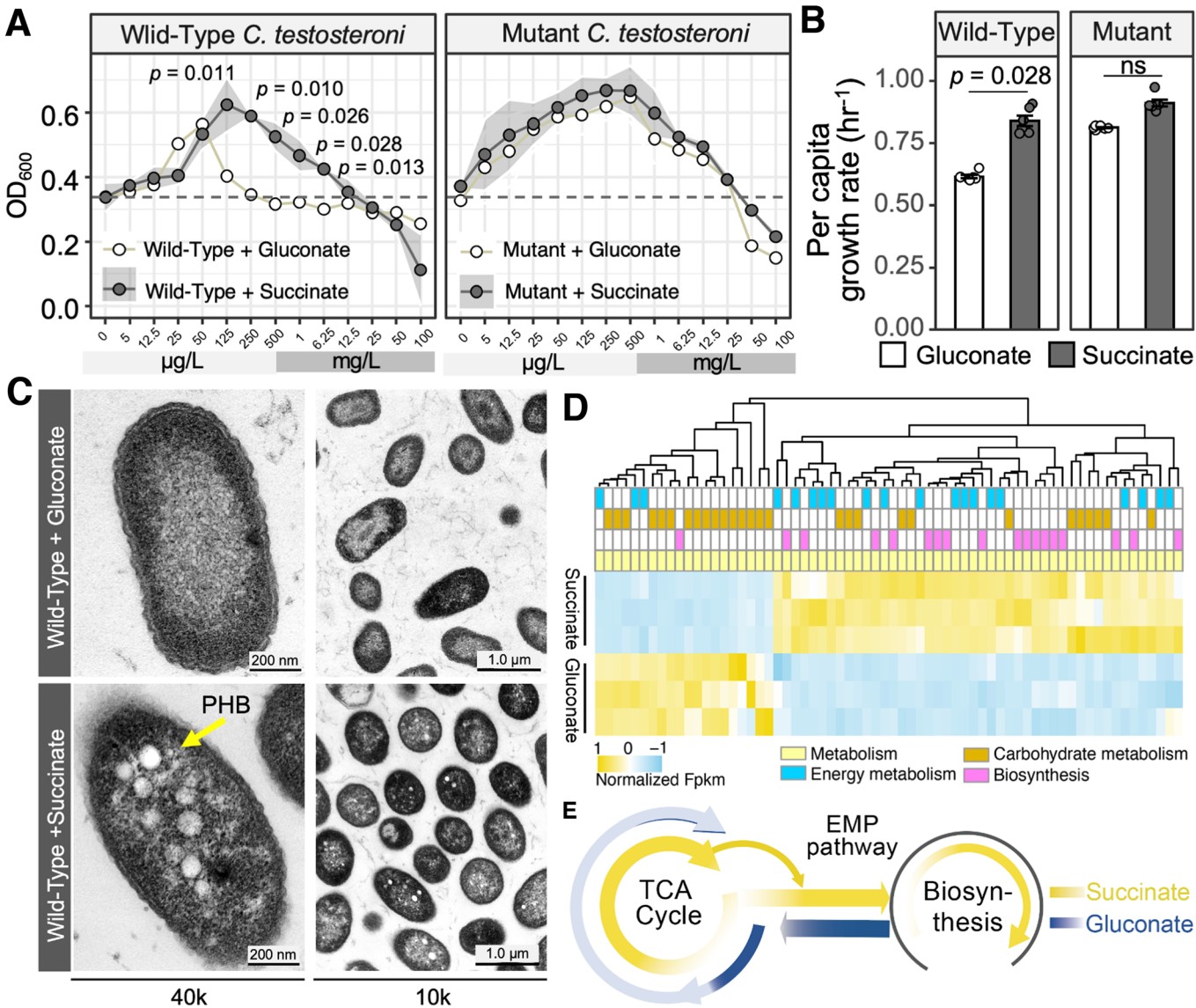

**Figure 5. Changes in metabolic patterns can independently initiate sulfamethoxazole (SMX) resistance in *Comamonas testosteroni*.**

(**A, B**) Examination of SMX resistance and per capita growth rates in wild-type and metabolically evolved *C. testosteroni* strains during cultivation on succinate and gluconate, highlighting differences between gluconeogenic and glycolytic pathways ($n = 6$). Growth data for cultures in succinate are normalized relative to those cultured in gluconate. The per capita growth rate referred to the population growth rate normalized by the initial population size. Growth rate data was expressed as mean ± cumulative standard deviation of six biological replicates. (**C**) TEM (scale bar: 1 μm) and zoomed-in TEM (scale bar: 200 nm) images of wild-type *C. testosteroni* cultured under succinate or gluconate. Yellow arrows represented poly-3-hydroxybutyrate (PHB). (**D**) Differential gene expression analysis in wild-type strains under SMX grown on succinate versus gluconate, presented through hierarchical clustering ($n = 3$). (**E**) Metabolic patterns of wild-type strains cultured on succinate and gluconate. Significance levels were all assessed using Wilcoxon test. Source data are available online for this figure.

2-ketogluconate or 6-phosphogluconate, integrating into the ED or PP pathways before entering the TCA cycle (Eisenberg and Dobrogosz, 1967).

During growth on media with equivalent carbon substrate concentrations, wild-type *C. testosteroni* demonstrated significantly higher growth rates (35% to 43%, Wilcoxon $P < 0.01$) and increased ZEP (100-fold, Wilcoxon $P < 0.01$) with succinate compared to gluconate (Appendix Table S6, Fig. 5A,B). This excessive carbon loss during gluconate catabolism led to a reduced biomass yield (by 68%) compared to succinate-grown cells (Wilcoxon $P < 0.01$). In

addition, TEM images of wild-type strains cultured with succinate revealed oval and round-shaped granular bodies akin to those observed in the dynamically evolved G250 isolates, indicating a similar metabolic regime leading to the accumulation of PHB (Fig. 5C). Finally, we explored the molecular basis of antibiotic resistance related to changes in metabolic pattern. Using RNA-seq, we compared gene expression differences between wild-type strains grown in gluconate versus succinate under SMX, revealing a significant transcriptional shift in 66 DEGs (Fig. 5D). Hierarchical clustering and KEGG pathway analysis distinguished two clear

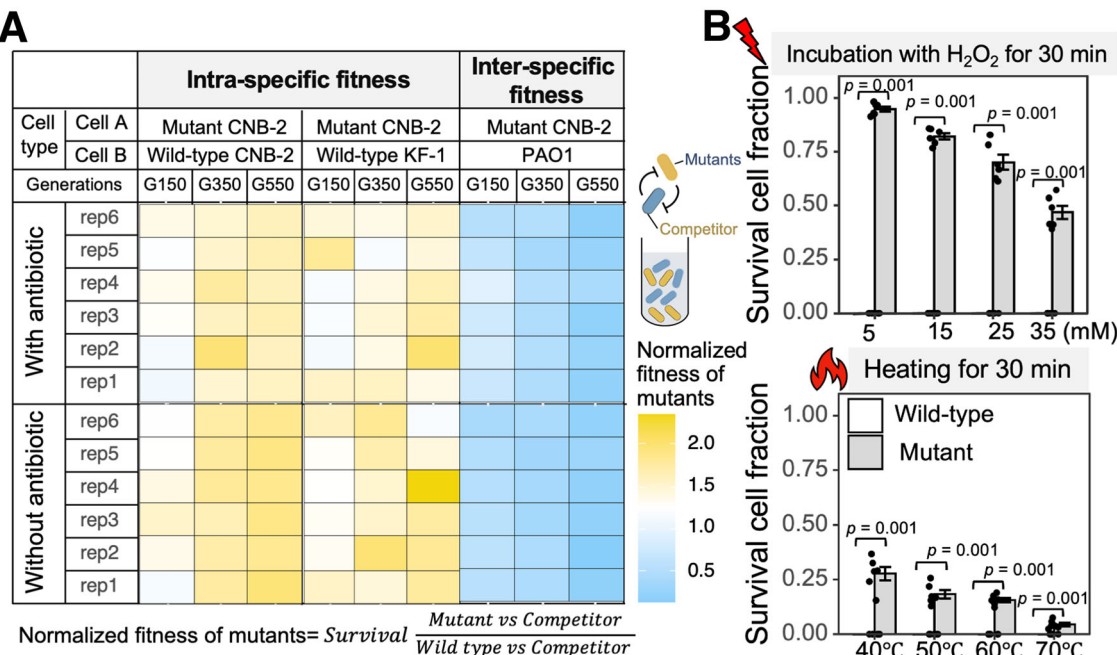

**Figure 6. Normalized growth fitness of evolved resistant isolates and cross-resistance.**

(**A**) Growth competition between evolved resistant isolates, wild-type *Comamonas testosteroni* CNB-2, kin bacteria *C. testosteroni* KF-1, and out-group species *Pseudomonas aeruginosa* PAO1 in NB medium with and without antibiotics. Six parallel populations containing cells A and B were performed at a 1:1 ratio. The heatmap displayed normalized fitness values calculated using the formula: Normalized fitness $= \frac{\text{fraction mutant vs. competitor}}{\text{fraction WT CNB-2 vs. competitor}}$ In this formula, "fraction" referred to the ratio of cell density, specifically measured as colony-forming units per milliliter (CFUs/mL), of each cocultured strain. When the competitor is the wild-type CNB-2, the normalized fitness simplified to: Normalized fitness = fraction mutant vs. WT CNB−2. (**B**) Evolved mutants were subjected to severe oxidative stress and heat shock to assess cross-resistance ($n = 6$). Survival data was expressed as mean ± cumulative standard deviation of six biological replicates. Significance levels were assessed using Wilcoxon test. Source data are available online for this figure.

phenotypes: an upregulation in biosynthetic metabolic genes ([FDR] *q*-value < 0.01) and a downregulation in core metabolic pathways ([FDR] *q*-value < 0.01), such as carbohydrate and energy metabolism, in succinate medium compared to gluconate medium (Fig. 5D).These findings suggested that SMX-resistant cells utilized metabolic regulation to enhance biomass production, helping to counteract the inhibitory effects of the biosynthesis-inhibiting antibiotic SMX, particularly on folate synthesis. This adaptive shift was driven by increased carbon flux through the TCA cycle, which ensured continued growth despite antibiotic pressure (Fig. 5E).

## Fitness evolutionary trajectories and cross-resistance in evolved variants

Having identified resistance mechanisms linked to metabolic strategies, we proceeded to evaluate the relative fitness of metabolically evolved variants. The evolutionary success of resistance mutants is significantly influenced by their relative fitness within a population. Specifically, mutants that exhibit enhanced fitness under selective pressure can outcompete wild types, leading to sustained phenotypic resistance. To assess this, we conducted competitive assays between metabolically evolved mutants and their wild-type counterparts, as well as with kin bacteria KF-1, using a 1:1 inoculum ratio. The chosen KF-1 strain of *C. testosteroni* shares 99% genetic identity with CNB-2, confirmed by 16S rRNA gene analysis (Ma et al, 2009). Similar

growth characteristics and antibiotic susceptibilities between KF-1 and CNB-2 enable accurate intra-specific competition dynamics analysis (Appendix Fig. S11). To quantify competitive fitness, the following formula was used:

$$\text{Normalized fitness} = \frac{\text{fraction mutant vs. competitor}}{\text{fraction WT CNB-2 vs. competitor}}$$

Values exceeding 1 (normalized fitness >1) signify enhanced fitness of the mutant strain relative to the wild type under competitive conditions, whereas values below 1 (0 < normalized fitness <1) indicate reduced fitness. Under selective SMX pressure, mutants dominated both wild-type CNB-2 and KF-1 populations, constituting about 100% of the population after 24 h, with a normalized fitness of 1.47 ± 0.22 (Fig. 6A; Appendix Figs. S12a and S13a,b).

However, the dominance of these resistance mutants was challenged in interspecific competitions with *Pseudomonas aeruginosa* PAO1. PAO1's robust resource acquisition abilities, extensive antimicrobial arsenal, and frequent coexistence with *C. testosteroni* in diverse habitats make *P. aeruginosa* an ideal model for assessing interspecific fitness dynamics under competitive conditions (Ghequire and De Mot, 2014; Lin et al, 2024; Sana et al, 2016). In competition experiments, the fraction of mutants significantly diminished to around 30%, with normalized fitness values of 0.62 ± 0.15 for G150, 0.43 ± 0.06 for G350, and 0.23 ± 0.08 for G550 (Fig. 6A; Appendix Figs. S12b and S13c, Wilcoxon *P* < 0.01). These findings suggested that while metabolic mutants maintain fitness

within their own species, this advantage did not persist in mixed-species environments and may even lead to potential extinction. To ensure that our findings were not influenced by pre-existing resistance mechanisms, we assessed the baseline susceptibility of PAO1 to SMX. With no prior exposure to the antibiotic, *P. aeruginosa* displayed typical growth inhibition across a range of SMX concentrations (Appendix Fig. S14a). Notably, the growth rate decreased under a concentration of 200 $\mu g \cdot L^{-1}$ SMX, demonstrating no significant resistance (Appendix Fig. S14b). This supported the notion that the observed decrease in interspecific fitness among resistance mutants was not due to antibiotic resistance in competing species but may be attributed to other, yet-to-be-identified, fitness costs. Furthermore, the same competition experiments conducted without selective pressure yielded similar competitive outcomes against both intra- and interspecies strains (Fig. 6A; Appendix Figs. S12 and S13). This suggested that evolutionary trajectories are not easily reversible by simply removing selective pressure.

Given the persistence of these evolutionary trends, we hypothesized that SMX resistance, induced by shifts in central metabolism, may also confer cross-resistance to various other stressors. To test, we exposed metabolically evolved variants to severe oxidative stress and heat shock. Results demonstrated that the mutants exhibited a significant increase in survival rates when subjected to extreme conditions, specifically heat shock (average survival rate of 73% ± 18% at temperatures ranging from 40 to 70 °C) and oxidative stress induced by $H_2O_2$ (average survival rate of 16% ± 9% at concentrations between 5 and 35 mM) (Fig. 6B). In contrast, the survival rates of wild-type cells under these conditions were nearly zero (Fig. 6B). Considering the influence of gluconeogenic catabolic pathways on resistance capacities, we supplemented succinate into the NB medium to assess wild-type cell survival under severe conditions (5 mM $H_2O_2$ oxidative stress and 40 °C heat shock). The addition of 100 mM succinate significantly improved the survival rates of wild-type cells to 34.6% ± 0.03% and 16.7% ± 0.02%, respectively (Appendix Fig. S15), further emphasizing the important role of gluconeogenic metabolic processes on antibiotic resistance.

# Discussion

Antibiotics are commonly seen as molecules that give their producers a competitive edge in ecological niches, diffusing from these cells into the extracellular environment (Wood et al, 2023). In structured microenvironments, they are more likely to encounter kin rather than adversaries, leading to dual roles where they benefit kin while harming foes—a dynamic that is highly concentration-dependent (Linares et al, 2006; Romero et al, 2011). At low doses, antibiotics can induce a range of non-lethal, yet significant effects, including extensive transcriptional reprogramming, stimulation of cell proliferation, activation of silent natural product biosynthesis gene clusters, and modulation of quorum sensing. These hormetic effects, which are critical stimulatory responses to low-dose stressors such as antibiotics, oxygen, metals, or nanomaterials, play a crucial role in the survival and adaptation of diverse biological entities, from microorganisms to plants and animals (Calabrese, 2008; Davies et al, 2006; Iavicoli et al, 2021). Despite the multifaceted roles of antibiotics, most resistance research has traditionally focused on their inhibitory effects, categorizing resistance

mechanisms into three main categories: target modification, drug inactivation, and drug transport (Blair et al, 2015; Woodford and Ellington, 2007). Among these, target modification is the most commonly recognized mechanism of bacterial resistance to sulfonamides. According to the Comprehensive Antibiotic Resistance Database (CARD), mutations in the conserved regions of the dihydropteroate synthase gene (*folP*) are the predominant sulfonamide resistance mechanism (Sköld, 2000). These are closely followed by the highly mobile sul genes (*sul1-sul4*) that encode DHPS enzymes not inhibited by sulfonamides, identified in the genetic material of the surveyed 88 pathogens (Nunes et al, 2020). These mutations, which have emerged across various bacterial lineages, are primarily chromosomal, although instances of horizontal gene transfer have been documented (Qvarnstrom and Swedberg, 2006). However, our extensive genetic analysis did not identify mutations in the genes that encode enzymes targeted by SMX, such as those within the *folP* and *sul* genes crucial for folate synthesis. By broadening our examination to include the potential stimulatory effects of antibiotics, our research proposes new genetic resistance mechanisms resulting from modifications and mutations in metabolic genes. This challenges traditional frameworks of resistance mechanisms, suggesting that existing categorizations may not fully capture all active mechanisms of action.

Indeed, metabolic adaptation may represent a distinct class of adaptive antibiotic resistance (adR) mechanisms, whereby cells alter their metabolic response to mitigate the downstream inhibition effects of antibiotics. Previous studies on the antibiotic-resistant metabolome of certain bacterial strains have revealed metabolic features that correlate with and contribute to resistance (Kuang et al, 2021; Su et al, 2021; Zhang et al, 2019). These findings suggested that it is possible to "reprogram" the antibiotic-resistant metabolome back to an antibiotic-sensitive state (Peng et al, 2015a; Peng et al, 2015b; Zhao et al, 2021). This is consistent with previous works in which metabolic constraints limit the evolution of antibiotic resistance (Zampieri et al, 2017; Zampieri et al, 2017). Furthermore, recent investigations have highlighted mutations in crucial areas, such as central carbon metabolism, energy generation, and biosynthesis, as recurring traits linked to antibiotic resistance in laboratory-evolved and clinical bacterial strains (Chen et al, 2022; Eoh et al, 2022; Lopatkin et al, 2021). However, detailed insights into how metabolic alterations contribute to stimulatory antibiotic resistance and the development of such resistance remain largely unexplored.

In this context, *C. testosteroni* provides a unique experimental model to explore the interplay between "beneficial" stress and metabolic adaptive evolution. In actively growing bacterial cells, metabolic fluxes are adapted to meet energy and redox demands (Shimizu and Matsuoka, 2019), with a trade-off between steady-state cell growth and physiological adaptability in fluctuating environments reported to be a conserved trait among bacteria (Basan et al, 2020). Our results supported the concept that widespread adjustments in metabolic fluxes accompany and enable resistance development (Bojanovič et al, 2017; Gottesman, 2019; Kim and Park, 2014; MacLean et al, 2020). Mechanistically, resistance evolved from highly unstable processes characterized by fluctuating methylation patterns in central metabolism, evolving into stable mechanisms driven by mutations in core metabolic genes. During this process, *C. testosteroni* optimized their carbon and energy fluxes to enhance growth capability, transitioning from glycolytic to gluconeogenic growth by (i) directly channeling

carbon into the TCA cycle and reducing metabolite secretion to meet elevated biosynthetic demands; (ii) reducing carbon and protein allocation to the EMP, oxidative PP, and ED pathways; and (iii) retaining carbon flux in the TCA cycle. Such metabolic remodeling may serve to minimize futile carbon cycling while favoring a fast-growing gluconeogenic metabolism.

Our research highlighted that evolving cell to serial exposure to increasing antibiotic concentrations led to heightened resistance and more intricate dynamics compared to those subjected to constant selective pressure. Under dynamically evolved protocols, the earlier emergence of instability in observed adR strongly suggested that it did not primarily involve DNA mutation events. Instead, the initial response of bacteria appears to involve rapid modulation of gene expression through epigenetic mechanisms. Over time, this strains exhibiting acquired resistance is continuously selected for and eventually evolves into inherent resistance. Similarly, prior studies have demonstrated that exposure to antibiotics can upregulate the expression of genes associated with efflux pumps (Fernández and Hancock, 2012; Olofsson and Cars, 2007). Such transient phenotypic changes are sometimes stabilized by DNA mutations that improve the efficiency and specificity of these efflux mechanisms (Sandoval-Motta and Aldana, 2016).This process marks a transition from an FTM, characterized by reversible phenotypic changes, to an SSM, which may overcome transient resistance through compensatory modifications (El Meouche and Dunlop, 2018), thus establishing enduring resistance to certain antibiotics. In our study, changes in DNA methylation patterns within the metabolic pathways eventually led to mutations in core metabolic genes. Furthermore, the rate of these mutations was also influenced by DNA methylation, suggesting that a rapid initial response to antibiotics could promote more stable genetic adaptations by enhancing genetic variability.

Our study also clearly distinguishes the acquired resistance we identified from the well-documented phenomenon of bacterial persisters. Research has shown that when E. coli K12 is exposed to a high concentration of ampicillin, only a small number of cells, approximately $10^5$ to $10^6$, survive as persisters (Moyed and Bertrand, 1983). These persisters temporarily cease growth in the presence of the antibiotic and resume only after its removal, without their progeny showing increased resistance. In contrast, the acquired resistance from our study did not show these typical persister traits such as dormancy, rarity, and stable progeny resistance. The distinct behavior of acquired resistance may be explained by the role of epigenetic memory mechanisms. Once gene expression patterns are established, these mechanisms help preserve these states across multiple generations. This transcriptional memory can be mediated through various processes, including DNA methylation patterns, as previously mentioned, inherited chromatin modifications, or features inherent to the genetic regulatory network (Owen et al, 2023). Consequently, in our acquired resistant isolates, resistance becomes a relatively permanent trait, maintained through these cellular memory mechanisms, contrasting with the transient resistance observed in persisters.

Furthermore, we observed that during metabolic evolution, increasing temperatures shifted the focus from growth adaptation to metabolic optimization, thereby accelerating the development of resistance and distinguishing specific metabolic variants. These observations support the growing body of evidence emphasizing the critical need to consider the relationship between global warming and microbial resistance trends for effective management of infectious diseases (MacFadden et al, 2018; Reverter et al, 2020; Rodríguez-Verdugo et al, 2020; Rzymski et al, 2024). Research conducted in the United States (MacFadden et al, 2018) and Europe (McGough et al, 2020) has linked higher average minimum temperatures with increased rates of antibiotic resistance. A potential mechanism for this trend is that elevated air temperatures may boost bacterial growth rates, thereby accelerating their evolutionary processes. Our study provides support for this theory, suggesting that warmer conditions may maximize metabolic adaptations and induce a hypermetabolic state in cells. The heightened metabolic activity could further accelerate mutations in core metabolic processes, favoring transitions towards biosynthetic metabolic pathways and hastening the emergence of antibiotic resistance. These findings underscore the urgent need to integrate research on bacterial evolution and global climate change, as rising temperatures may worsen antibiotic resistance, intensifying the challenge of combating infectious diseases.

This urgency is underscored by the clinical challenges posed by infections from C. testosteroni. Known for its prevalence in diverse environments, the broad adaptability of C. testosteroni makes it a potential agent for mild yet persistent infection in clinical settings (Farooq et al, 2017; Farshad et al, 2012). This pathogen has been implicated in a wide array of conditions, from cellulitis to more severe infections such as peritonitis, endocarditis, and bacteremia (Orsini et al, 2014). A major obstacle in treatment is the lack of established antibiotic susceptibility breakpoints for C. testosteroni in EUCAST guidelines. This absence necessitates reliance on broader-spectrum or empirical antibiotic strategies. These may not be optimal for treating C. testosteroni, which responds uniquely to the pressure of antibiotics. In addition, the gradual decay of antibiotics in patients post-treatment creates an inconsistent concentration gradient, further complicating the effective management of these infections. Advancing research to uncover novel resistance mechanisms could lead to more targeted and effective treatment strategies, which are crucial for controlling the spread and evolution of antibiotic resistance in such atypical pathogens.

# Methods

**Reagents and tools table**

| Reagent/resource | Reference or source | Identifier or catalog number |
|---|---|---|
| **Experimental models** | | |
| *Comamonas testosteroni* | Horinouchi et al, 2019 | CNB-2 (ATCC 11966) |
| | Weiss et al, 2013 | KF-1 (NBRC 100989) |
| *Escherichia coli* | Takara Bio | β2155 |
| *Pseudomonas aeruginosa* | ATCC 15692 | PAO1 |
| **Recombinant DNA** | | |
| pCVD442 | Sangon | |
| **Antibodies** | | |
| **Oligonucleotides and other sequence-based reagents** | | |
| RT-PCR primers | This study | Appendix Table S4 |

| Reagent/resource | Reference or source | Identifier or catalog number |
|---|---|---|
| Site-directed mutagenesis primers | This study | Appendix Table S6 |
| **Chemicals, enzymes, and other reagents** | | |
| Sulfamethoxazole | Sigma | S7507 |
| Gentamicin | Sigma | E003632 |
| CN-agar plates | Hopebio | HB84842 |
| DNA ligase | Thermo Scientific | B300056 |
| SmaI | Thermo Scientific | B300191 |
| Buffer Tango | Thermo Scientific | B300049 |
| PrimeSTAR Max Premix | TaKaRa | R045Q |
| cDNA synthesis kit | Qiagen | 205311 |
| **Software** | | |
| R v4.3.3 | https://www.r-project.org | |
| gplots R package v3.1 | https://cran.r-project.org/web/packages/gplots/ | |
| MATLAB (R2018a) | https://www.mathworks.com/?s_tid=gn_logo | |
| **Other** | | |
| PacBio RS II instrument | Pacific Bioscience | |
| NovaSeq PE150 | Illumina | |
| Novaseq 6000 | Illumina | |
| Leica UC7 ultramicrotome | Leica Microsystems | |
| Hitachi HT7800 TEM | Hitachi High-Technologies Corporation | |
| Thermo Scientific Multiskan FC | Thermo Scientific | |
| Qubit® 2.0 Fluorometer | | |
| LightCycler 96 Real-Time PCR system | Roche Diagnostics | |
| QTRAP® 6500 LC-MS/MS System | SCIEX | |

## Bacterial strains and evolution protocols

All bacterial strains, gene mutants, and plasmids used in this study are listed in Appendix Table S7. The evolution experiments used the *C. testosteroni* CNB-2 (ATCC 11966) strain, whose cell growth is promoted by low doses of SMX ($< 250\ \mu g \cdot L^{-1}$) (Lin et al, 2023). Briefly, a single colony was picked from an agar plate and grown in 5 mL of nutrient broth (NB) medium overnight with shaking at 30 °C/170 rpm. After 12 h, cells were diluted 50× into fresh NB medium and aliquoted into one row (four wells) of a 12-well plate for each evolution. For static and dynamic evolution, after the first cycle of regrowth, cells were diluted 50× into fresh rich medium daily; antibiotics were added to each treatment well, while sterile water was added to the control at the concentrations shown in Appendix Table S8. The protocol for metabolic evolution was identical to that for static evolution, with the following modifications. During the evolutionary process, the incubation temperature gradually increased

by 1 °C every three days from 18 °C, culminating at 37 °C on day 55. All other experimental parameters are detailed in Appendix Table S8. All evolution protocols were conducted for about 550 generations (55 days). The cultures were preserved in 50% glycerol and stored at −80 °C every 50 generations. For all evolution protocols, optical density at 600 nm ($OD_{600}$) was measured before daily dilution using a microplate reader (Thermo Scientific Multiskan FC, Vantaa, Finland). In addition to the NB medium utilized for evolutionary experiments, gluconate-mineral salt medium (MSM) and succinate-MSM were used to culture wild-type strains. This setup was intended to determine if wild-type strains could directly modify their metabolic strategies to enhance resistance by utilizing different carbon sources. Details of the culture media are presented in Appendix Table S9. In addition, no blinding was applied in this study, as it did not involve human participants. The experimental conditions (e.g., treatment or control) were known during data collection and analysis.

## MIC test and ZEP calculation of evolved strains

The evolved strains were subjected to MIC tests every two days to determine their phenotypic tolerance levels. The cell culture was diluted with PBS to an $OD_{600}$ of 0.1, regarded as the standard solution. Subsequently, 2 μL of standard solution was added to fresh NB medium (100 μL) containing SMX at a series of concentrations. A growth control without antibiotics and a negative control without bacterial inoculum were also established. Cell cultures were incubated at 30 °C for 24 h, followed by $OD_{600}$ measurement. The ZEP is defined as the highest antibiotic dose at which the CRC intersects the control-response line, indicating the maximum dose that *C. testosteroni* can tolerate without significant adverse effects (Dorato and Engelhardt, 2005). The ZEP was determined by analyzing CRCs, which show the growth response of *C. testosteroni* to varying concentrations of SMX. Data were fitted using a sigmoidal model employing the Hill inverse dichotomy iteration technique, enabling precise identification of the transition point. The model parameters, including the ZEP, were optimized using the Levenberg-Marquardt algorithm, as described previously (Wang et al, 2018), enhancing the accuracy and reliability of our findings. Details are provided in the Appendix Method S1.

## Measurement of growth rates

Growth rate assessments for the evolved strains were conducted every day under various evolutionary protocols, including control, static, dynamic, and metabolic evolution. Cell cultures were initiated at an $OD_{600}$ of 0.1 and incubated in 96-well NB plates under SMX-free conditions. The plates were shaken at 170 rpm and maintained at a constant temperature of 30 °C. The plates were tightly sealed, and OD was monitored at a wavelength of 600 nm. Growth rates were measured by assuming exponential growth to a threshold of OD 0.5 and the effective growth rate was determined by calculating the ratio $\lg(OD_{threshold}/OD_i)/T_{threshold}$, as previously described (Lin et al, 2024). This time-to-threshold measurement accounts for lag times, as strains with a time lag will reach the threshold OD later than those without, despite having similar exponential growth rates. In addition, this method was used to assess the growth rates of evolved strains at specific time points—days 15, 25, 35, and 55—across a range of SMX concentrations from 0.005 to 100 mg·L$^{-1}$.

## Genomic DNA extraction, library prep, and high-throughput sequencing

To identify genetic changes in the evolutionary process, mutants in the evolved populations were isolated using SMX-containing agar at days 15, 25, 35, and 55 under static, dynamic, and metabolic evolution, with three replicates per treatment. Genomic DNA was extracted using the sodium dodecyl-sulfate (SDS) method, with the quality of the harvested DNA assessed using a Qubit® 2.0 Fluorometer (Thermo Scientific). Extracted DNA was subjected to PacBio sequencing and whole-genome sequencing. Details are provided in Appendix Method S2.

## Quantitative real-time polymerase chain reaction (qRT-PCR)

The differential expression levels of core metabolic genes (*AcnB*, *TctA*, *Bug*, *ALDH*, *GltA*, *NadB*, *Pck*) in G150, G350, G550 evolved isolates were validated using qRT-PCR. For reverse transcription analysis, RNA samples were used as templates to synthesize cDNA using a cDNA synthesis kit (Qiagen, Germany). The resultant cDNA was then used for qRT-PCR using the LightCycler 96 Real-Time PCR system (Roche Diagnostics, Switzerland). The relative expression level (copy number of mRNA transcripts) of each target gene was normalized to the cDNA concentration. All primer pairs used for qRT-PCR analysis are listed in Appendix Table S10.

## Targeted metabolomics analysis of central carbon metabolism with liquid chromatography-tandem mass spectrometry (LC-MS/MS)

After 24 h of growth, G550 cell cultures evolved with or without SMX stress were collected, freeze-dried, and reconstituted in water. Subsequently, 100 μL of each sample was homogenized with 500 μL of methanol/water (8:2) containing mixed internal standards. After 30 min of incubation on ice, the mixture was centrifuged at 12,000 rpm for 10 min. The resulting supernatant was injected into an ExionLC™ AD UHPLC-QTRAP 6500+ system for analysis. The UHPLC-MS/MS system, operated by Novogene Co., Ltd. (Beijing, China), utilized a Waters Atlantis Premier BEH Z-HILIC column (2.1 × 100 mm, 1.7 μm) maintained at 50 °C. The mobile phase, composed of 15 mM ammonium acetate with 10 mM imino-bis (methyl phosphonic acid) (solvent A) and 15 mM ammonium acetate/acetonitrile (solvent B), was delivered at a flow rate of 0.40 mL/min. Appendix Table S11 provides information on categories and MS detection parameters for all central carbon metabolism substances tested. Energy charge was calculated as described previously (Chapman et al, 1971; Kayser et al, 2005) from the quantified pools of adenosine nucleotides using the following formula:

$$\frac{ATP + 0.5[ADP]}{[ATP] + [ADP] + [AMP]}$$

## Mutation validation

Genes with known SNPs—*TctA*, *Pck*, *NadB*, *GltA*, *ALDH*, and *AcnB*—identified in metabolically evolved *C. testosteroni* mutants were targeted for reversion to their ancestral states. High-fidelity DNA polymerase was used to amplify the upstream and downstream sequences of each mutation site, which were fused via PCR to create homologous recombination arms. These fused arms were cloned into the suicide plasmid pCVD442GS, and the resulting plasmids were electroporated into *E. coli* β2155 to create donor strains for each mutant gene. The donor strains were then used in conjugation experiments with metabolically evolved *C. testosteroni* mutants, and 50 μL recombinants were selected on LB plates 30 μg·mL⁻¹ containing gentamicin. These colonies represented single-crossover recombinant clones that integrated the complete suicide plasmid via homologous recombination at one arm, referred to as CNB-2/pCVD442GS-mutant. Following further culturing, double-crossover recombinant clones were selected, resulting from a secondary homologous recombination event that either introduced the desired mutation in the target gene or reverted the strain to its original form. To remove single-crossover clones, selected colonies from the gentamicin-resistant plates were streaked onto LB plates containing 10% sucrose and incubated overnight at 37 °C/220 rpm. The clones were PCR-verified and sequenced to confirm the correct mutation. After three rounds of streak plating, the final mutant strain was confirmed. Appendix Tables S7 and S12 list the strains, plasmids, and primers used in this study. All experiments were performed with 30 μg·mL⁻¹ gentamicin, and MICs were measured as previously described.

## Statistical analysis

Significance (*$P < 0.05$) was determined using the non-parametric Wilcoxon test. All statistical analyses were performed in R v3.1.2 (R Development Core Team, 2015). Figures were prepared using the basic R package and ggplot2.

# Data availability

The datasets produced in this study are available in the following databases: Whole-genome sequencing: NCBI GEO Short Read Archive (SRA) PRJNA1103939. Single-molecule real-time sequencing (PacBio): NCBI GEO Short Read Archive (SRA) PRJNA1104141.

The source data of this paper are collected in the following database record: biostudies:S-SCDT-10_1038-S44320-025-00087-4.

# Peer review information

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

## Acknowledgements

This work was supported by the National Natural Science Foundation of China (52388101 and 52250056).

## Author contributions

**Hui Lin**: Conceptualization; Data curation; Formal analysis; Investigation; Visualization; Methodology; Writing—original draft; Writing—review and editing. **Donglin Wang**: Conceptualization; Investigation; Methodology; Writing—review and editing. **Qiaojuan Wang**: Conceptualization; Investigation; Writing—review and editing. **Jie Mao**: Investigation; Writing—review and editing. **Lutong Yang**: Writing—review and editing. **Yaohui Bai**: Supervision; Funding acquisition; Validation; Visualization; Writing—review and editing. **Jiuhui Qu**: Funding acquisition.

Source data underlying figure panels in this paper may have individual authorship assigned. Where available, figure panel/source data authorship is listed in the following database record: biostudies:S-SCDT-10_1038-S44320-025-00087-4.

## Disclosure and competing interests statement

The authors declare no competing interests.

# Expanded View Figures

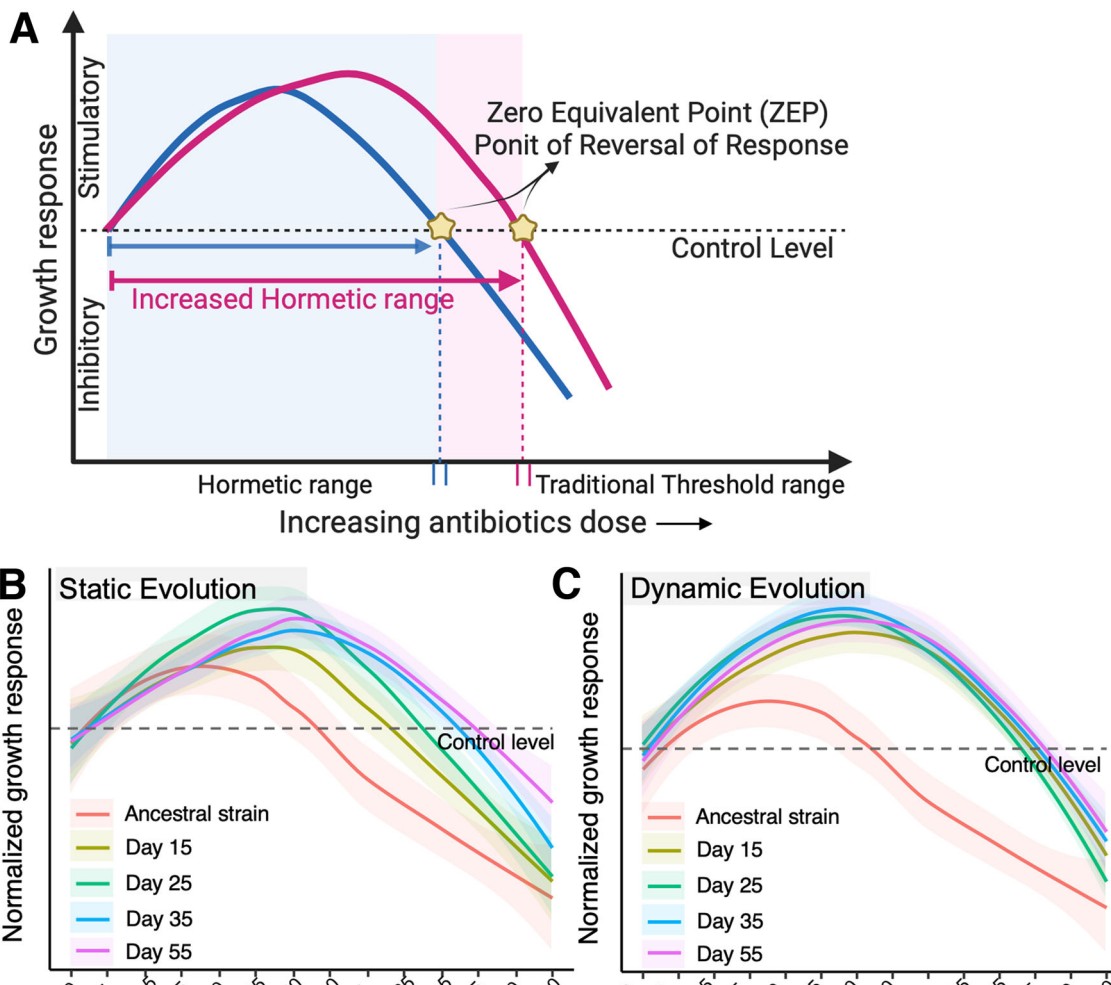

**Figure EV1.  Concentration-response curves (CRCs) for *Comamonas testosteroni* exposed to sulfamethoxazole (SMX).**

(**A**) Schematic representation of a CRC. Fitted CRCs of *C. testosteroni* during (**B**) static evolution and (**C**) dynamic evolution at different time points: days 15 (G150), 25 (G250), 35 (G350), and 55 (G550). Growth data for the evolved isolates were normalized relative to the ancestral strain in an SMX-free environment, utilizing data from six biological replicates.

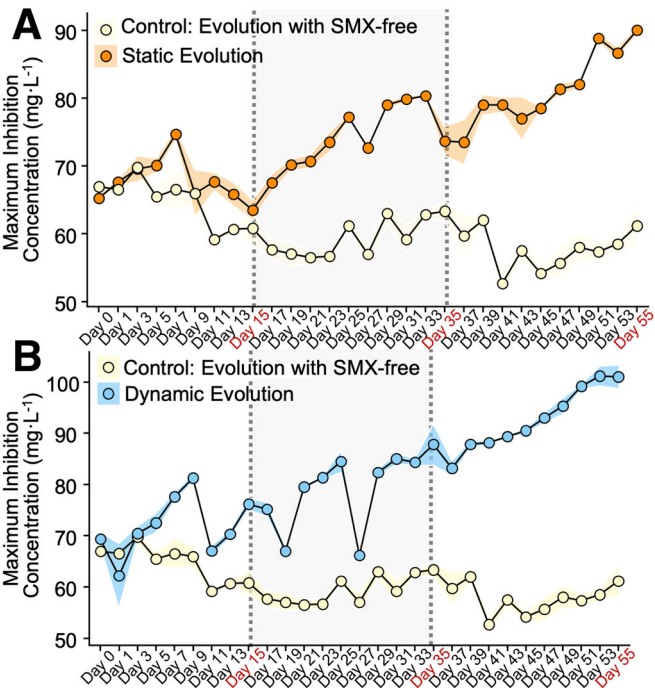

**Figure EV2. Minimum Inhibitory Concentration (MIC) patterns observed over 550 generations of *Comamonas testosteroni* evolution under static and dynamic evolution protocols.**

(A) Static evolution protocol; (B) Dynamic evolution protocol. Each evolutionary scheme comprised three biological replicates.

   

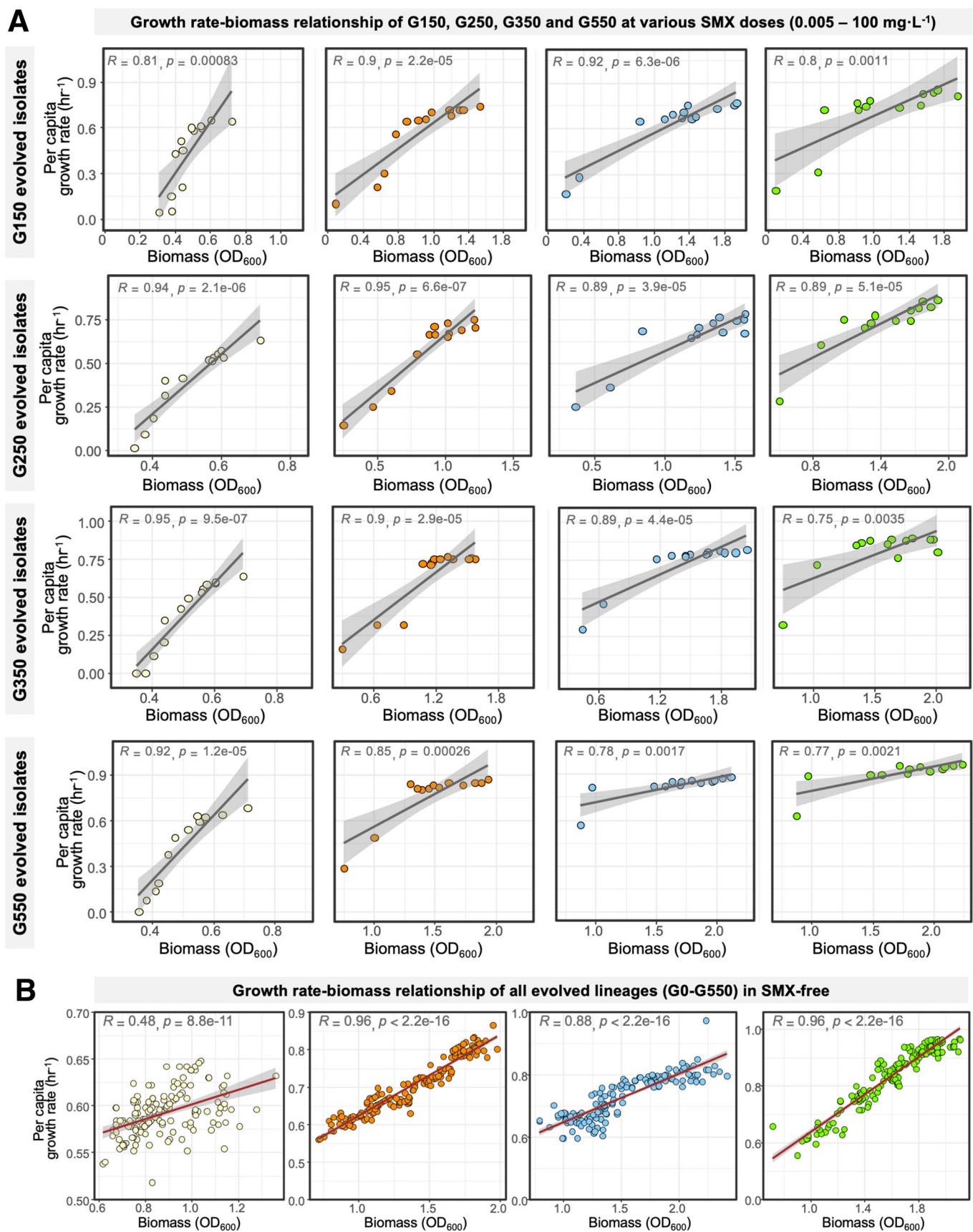

**A** Growth rate-biomass relationship of G150, G250, G350 and G550 at various SMX doses (0.005 – 100 mg·L$^{-1}$)

**B** Growth rate-biomass relationship of all evolved lineages (G0-G550) in SMX-free

◀ **Figure EV3.  Correlation between growth rate and biomass (OD$_{600}$) in evolved strains.**

(**A**) Across a range of sulfamethoxazole (SMX) concentrations (0.005 to 100 mg·L$^{-1}$) in the evolved isolates G150, G250, G350, and G550; (**B**) Across all evolved strains (G0-G550, control, static, dynamic, and metabolic evolution) in SMX-free environments. Each panel included the regression line, Pearson correlation coefficient (R), and the *P* value from the *T* test, indicating the statistical significance of the correlations.

  