## [Peer Review File · Molecular Systems Biology]

Epigenetic Modifications and Metabolic Gene Mutations Drive Resistance Evolution in Response to Stimulatory Antibiotics

Hui Lin, Donglin Wang, Qiaojuan Wang, Jie Mao, Lutong Yang, Yaohui Bai, and Jiuhi Qu

Corresponding author: Yaohui Bai (yhbai@rcees.ac.cn)

Review Timeline:

Submission Date:	30th Jul 24
Editorial Decision:	13th Sep 24
Revision Received:	29th Nov 24
Editorial Decision:	23rd Dec 24
Revision Received:	29th Dec 24
Accepted:	7th Jan 25

Editor: Poonam Bheda

Transaction Report:

13th Sep 2024

Manuscript Number: MSB-2024-12544

Title: Epigenetic Modifications and Metabolic Gene Mutations Drive Resistance Evolution in Response to Stimulatory Antibiotics

Dear Dr. Bai,

Thank you for the submission of your manuscript to Molecular Systems Biology. We have now received feedback from the two reviewers who agreed to evaluate your manuscript. As you will see from the reports below, the referees acknowledge the interest of the study and are overall supportive of your work; however they also comment on multiple aspects of the manuscript that should be strengthened in a revision.

I think that the recommendations of the reviewers are rather clear and I therefore do not see the need to repeat the comments listed below. One of the more fundamental points raised refers to claims that metabolic changes are driving antibiotic resistance, which may be correlated/consequence. We would encourage you to address this point fully in your revision. All other issues raised would need to be satisfactorily addressed. Please let me know in case you would like to discuss in further detail any of the issues raised, I would be happy to schedule a call.

We require:

- 1) A .docx formatted version of the manuscript text (including legends for main figures, EV figures and tables). Please make sure that the changes are highlighted to be clearly visible. Alternatively you may choose to submit your manuscript as a LaTeX file.
- 2) Individual production quality figure files as .eps, .tif, .jpg (one file per figure). For guidance, download the 'Figure Guide PDF' (<https://www.embopress.org/page/journal/17574684/authorguide#figureformat>).
- 3) At EMBO Press we ask authors to provide source data for the main figures. Our source data coordinator will contact you to discuss which figure panels we would need source data for and will also provide you with helpful tips on how to upload and organize the files.
- 4) A .docx formatted letter INCLUDING the reviewers' reports and your detailed point-by-point responses to their comments. As part of the EMBO Press transparent editorial process, the point-by-point response is part of the Peer Review File (PRF), which will be published alongside your paper.
- 5) A complete author checklist, which you can download from our author guidelines (<https://www.embopress.org/page/journal/17574684/authorguide#submissionofrevisions>). Please insert information in the checklist that is also reflected in the manuscript. The completed author checklist will also be part of the PRF.
- 6) Please note that all corresponding authors are required to supply an ORCID ID for their name upon submission of a revised manuscript.
- 7) It is mandatory to include a 'Data Availability' section after the Materials and Methods. Before submitting your revision, primary datasets produced in this study need to be deposited in an appropriate public database, and the accession numbers and database listed under 'Data Availability'. Please remember to provide a reviewer password if the datasets are not yet public (see <https://www.embopress.org/page/journal/17574684/authorguide#dataavailability>).

This study includes no data deposited in external repositories.

- 8) All Materials and Methods need to be described in the main text using our 'Structured Methods' format, which is required for all research articles. According to this format, the Methods section includes a Reagents and Tools Table (listing key reagents, experimental models, software and relevant equipment and including their sources and relevant identifiers) followed by a Methods and Protocols section describing the methods using a step-by-step protocol format. The aim is to facilitate adoption of the methodologies across labs. Please upload the Reagents and Tools table as a separate document when submitting your revised manuscript. More information on how to adhere to this format as well as a downloadable template (.docx) for the Reagents and Tools Table can be found in our author guidelines:

<https://www.embopress.org/page/journal/17444292/authorguide#structuredmethods>

An example of a Method paper with Structured Methods can be found here:
<https://www.embopress.org/doi/10.15252/msb.20178071>.

9) For data quantification: please specify the name of the statistical test used to generate error bars and P values, the number (n) of independent experiments (specify technical or biological replicates) underlying each data point and the test used to calculate p-values in each figure legend. The figure legends should contain a basic description of n, P and the test applied. Graphs must include a description of the bars and the error bars (s.d., s.e.m.). Please provide exact p values.

10) Our journal encourages inclusion of *data citations in the reference list* to directly cite datasets that were re-used and obtained from public databases. Data citations in the article text are distinct from normal bibliographical citations and should directly link to the database records from which the data can be accessed. In the main text, data citations are formatted as follows: "Data ref: Smith et al, 2001" or "Data ref: NCBI Sequence Read Archive PRJNA342805, 2017". In the Reference list, data citations must be labeled with "[DATASET]". A data reference must provide the database name, accession number/identifiers and a resolvable link to the landing page from which the data can be accessed at the end of the reference. Further instructions are available at .

11) We replaced Supplementary Information with Expanded View (EV) Figures and Tables that are collapsible/expandable online. A maximum of 5 EV Figures can be typeset. EV Figures should be cited as 'Figure EV1, Figure EV2' etc... in the text and their respective legends should be included in the main text after the legends of regular figures.

<https://www.embopress.org/page/journal/17574684/authorguide#expandedview>

13) Author contributions: CRediT has replaced the traditional author contributions section because it offers a systematic machine readable author contributions format that allows for more effective research assessment. Please remove the Authors Contributions from the manuscript and use the free text boxes beneath each contributing author's name in our system to add specific details on the author's contribution. More information is available in our guide to authors.

Please also suggest a striking image or visual abstract to illustrate your article as a PNG file 550 px wide x 300-600 px high. Share synopsis text and image, as well as eTOC:

Please note that these would be the final versions and changes during proofing are usually not allowed

16) As part of the EMBO Publications transparent editorial process initiative (see our policy here:

https://www.embopress.org/transparent-process#Review_Process), Molecular Systems Biology will publish online a Peer Review File (PRF) to accompany accepted manuscripts.

In the event of acceptance, this file will be published in conjunction with your paper and will include the anonymous referee reports, your point-by-point response and all pertinent correspondence relating to the manuscript. Let us know whether you agree with the publication of the PRF and as here, if you want to remove or not any figures from it prior to publication.

Please note that the Authors checklist will be published at the end of the PRF.

Molecular Systems Biology has a "scooping protection" policy, whereby similar findings that are published by others during review or revision are not a criterion for rejection. Should you decide to submit a revised version, I do ask that you get in touch

after three months if you have not completed it, to update us on the status.

I look forward to receiving your revised manuscript.

Yours sincerely,

Poonam Bheda, PhD
Scientific Editor
Molecular Systems Biology

Reviewer #1:

Hui Lin et al investigate the evolution of antibiotic resistance in *Comamonas testosteroni* when exposed to sub-MIC concentrations of sulfamethoxazole (SMX). The authors demonstrate that under these conditions, resistance evolves in two ways: a fast, reversible mechanism driven by DNA methylation of genes in central metabolic pathways, and a slower, stable mechanism involving mutations in metabolic genes. These changes lead to a metabolic shift from glycolysis to gluconeogenesis, which enhances bacterial growth and biomass production. The study also shows that increasing temperatures accelerate the evolution of resistance by promoting mutations in metabolic genes. The findings suggest that metabolic adaptations are relevant for antibiotic resistance, but it is difficult to distinguish cause and consequence from the data. Moreover, the role of global growth rate effects on both metabolic changes and antibiotic resistances is unclear. Thus, although the authors show interesting data and the role of metabolism for antibiotic action is a relevant topic, some points should be clarified to support the claims:

Main points:

- 1) The authors use the zero effective concentration point ZEP as a measure of resistance, but it remains unclear how ZEP directly relates to the MIC and why ZEP is a more informative measure. The authors should clarify the rationale behind using ZEP over MIC and provide data to support the claim that ZEP has better insights into resistance mechanisms. Otherwise replace ZEP with MIC, which is a common measure for resistance also in the clinic. Also the authors should include how SMX levels relate to EUCAST breakpoints to understand clinical relevance.
- 2) The metabolic changes and also the stimulatory effect, could potentially be caused by reduced bacterial growth rates under sub-MIC conditions rather than the direct effects of the antibiotics or the mutations. Antibiotic-based reduction of growth rates are known to strongly influence metabolism and proteome allocation (e.g. work of Terry Hwa DOI: 10.1126/science.1192588). The authors should describe how their gene expression and metabolism changes relate to such global growth effects. Further reduced growth rates are known to affect antibiotic susceptibility, and this could confound the conclusions. The authors need to report and analyze growth rates across all experimental conditions to rule out such general growth rate effects.
- 3) Related to point 3, the authors claim that better growth comes from more efficient resource allocation (e.g. line 260). However, as I understand they measure growth based on end ODs after 24 h. But this can be achieved by reduced growth rates that would lead to higher end ODs because it reduces overflow metabolism. The authors need quantitative data about physiologies, at least biomass specific growth rates to make such claims about efficiency of metabolism.
- 4) The authors claim that metabolic changes are a driving force behind the emergence of resistance. For example in the abstract "new genetic mechanisms of resistance linked to metabolic remodeling". However, the metabolic changes may also just be the consequence rather than the cause of resistance. The authors should provide additional evidence to support the claim that the metabolic changes are causal for resistance and not merely downstream effects of other resistance mechanisms. The most direct evidence would be to insert one of the metabolic mutations individually, although I understand that one mutation alone could be insufficient. Maybe another approach to show causality is to revert mutations in an evolved isolate to illustrate that they contribute directly to resistance.
- 5) Related to 4, did the authors find known SMX resistance mutations e.g. in the target gene or in transporters? To authors should at least better discuss the role of canonical resistance mechanism in their experiments.

Minor:

In general the figure captions and method section should be better described to understand what data is shown. For example Figure 1d: please define per capita growth rate

According to Figure 1 and Figure S1 the authors measured MICs. Also, in Figure 3a it says MIC of n= 3 replicates but it is not clear which data this is. This is confusing and should be clarified.

Reviewer #2:

Title: Epigenetic Modifications and Metabolic Gene Mutations Drive Resistance Evolution in Response to Stimulatory Antibiotics
Summary:

This study examined the evolution of antibiotic resistance under a stimulatory antibiotic. Experimental evolution of resistance to sulfamethoxazole in *C. testosteroni* CNB-2, whose growth is promoted by low doses of sulfamethoxazole (SMX), a fact found by a previous study by this group was performed. Populations taken at different stages of the evolution experiment were sequenced to trace SNPs that may contribute to SMX resistance. Once the evolution experiment was completed resistant isolates were subjected to a wide variety of analysis (MIC/ZEP determination, qRT-PCR, competition assays, growth assays, TEM, heat and oxidative stress assays, RNA-seq, and LC-MS/MS). All together this study suggests and supports novel mechanisms of resistance linked to metabolic remodeling for this system. There are two mechanisms a fast epigenetic response (though metalation) and a slower selection for mutations both of which occur in pathways related to metabolism. Both mechanism contribute to a transition from glycolysis to gluconeogenesis. The study also shows how an increase in environmental temperature can also accelerate the metabolic evolution by increasing the number of mutations in metabolic genes. However, these mutants are only favored in environments treated with SMX and are nearly or completely competitively excluded in both intra- and inter- specific co-cultures without SMX. This work adds to bring light to the growing need for environmental factors, especially those related to climate change, to be taken into greater consideration when examining the antibiotic resistance crisis.

General remarks

Overall, the key conclusions are well supported. I hope that the trends from the competition assays remain once the WT CNB-2 experiments are performed (see the last major point for more details). This study is a perfect example of the need for more interdisciplinary approaches to the antibiotic resistance problem from an ecological and evolutionary standpoint. This unique study system allows for the authors to thoroughly investigate a previously overlooked aspect of a concentration response curve and shows novel antibiotic resistance mechanisms involving metabolism. The study will attract the interest of evolutionary, ecological, and clinical microbiologist along with those interested in public health.

Major points

- Lines: 68-80 - It would have been great to see how tolerance and persistence plays into these dynamics. Persistence is mentioned in the discussion, but it would be good to include it in the introduction.
- Lines: 109-110 - For the metabolic evolution experiment what is the rationale for the temperature range used? Were these temperatures the fundamental niche width of the thermal performance curve of the WT/ancestral strain?
- Lines: 115-117 - It is unclear in the methods if the experimental evolution occurred with the MSM or succinate-MSM medium. After reading the results it was made clear when this medium was used but it would have been beneficial to have this more explicitly stated in the methods.
- Lines: 118-127 - Although references were provided from a previous study looking at the ZEP it would be great to have a brief summary of how the ZEP was determined and the definition of the ZEP.
- Figure 1d - How were growth rates determined?
- Line: 472 - What was the rationale behind the species selection for interspecies competition assay? Did PAO1 have previous exposure to SMX? Did this strain of PAO1 recently evolve antibiotic resistance? Depending on this answer there might be compensatory mutations to maintain better growth thus making it a better competitor when your stain has not had this time yet. It would be good to know what the competition assays are between WT CNB-2 v KF-1 and WT CNB-2 v WT PAO1 to have a point of comparison. Then the results can be normalized to the WT CNB-2 results to see how the mutations may have affected the ability to compete in coculture.

Minor points

- Line: 164 - At times the results section reads more like a discussion. This is purely a stylistic choice, but I would try and limit speculation/suggestions in the results section and keep it to the discussion section.
 - Example: Lines 192-196
- Line: 180 - typo. "resistant" should be "resistance"
- Figure 3d - It would have been nice if the color scheme was the same as Figure 3a & 3c
- Line: 468 - What was the rationale for the strain selected in the competition assays? Are there any phenotypic differences between KF-1 and the WT CNB-2?
- Line: 531 - Are there any other known stimulatory effects observed between an antibiotic and bacterial species? It would be good to touch on this to help give additional relevance to the resistance mechanisms that are brought to light here.
- Lines: 586-609 - I would perhaps separate this out to two different paragraphs. I would also suggest including the following citations:

- o Rodríguez-Verdugo, A., Lozano-Huntelman, N., Cruz-Loya, M., Savage, V. and Yeh, P., 2020. Compounding effects of climate warming and antibiotic resistance. *IScience*, 23(4).
- o Reverter, M., Sarter, S., Caruso, D., Avarre, J.C., Combe, M., Peppey, E., Pouyaud, L., Vega-Heredía, S., De Verdal, H. and Gozlan, R.E., 2020. Aquaculture at the crossroads of global warming and antimicrobial resistance. *Nature communications*, 11(1), p.1870.
- o Rzymiski, P., Gwenzi, W., Poniedziałek, B., Mangul, S. and Fal, A., 2024. Climate warming, environmental degradation and pollution as drivers of antibiotic resistance. *Environmental Pollution*, p.123649.

Response to Reviewers' comments:

We are grateful to the editor and reviewers for their detailed evaluations and constructive feedback. In response, we have updated our data, conducted additional experiments, revised the scheme for better clarity, and made substantial revisions to the manuscript.

General Revisions:

- We clarified the definition of the Zero Effective Concentration Point (ZEP) and detailed its significance as a resistance measure in our study context.
- We systematically documented growth rates across all experimental conditions, detailed in Appendix Table S1, Figure EV3, S2, and S4, and added a “*Measurement of Growth Rates*” subsection in the Materials and Methods.
- We expanded the Introduction to discuss tolerance and persistence behaviors and enriched the Discussion section to emphasize the clinical relevance of our findings.

Specific Responses:

- **Reviewer 1:** We performed reverse mutation experiments on metabolically evolved mutants to substantiate that these mutations contribute directly to resistance, not merely as secondary effects. Detailed methodologies for these procedures and information on mutation loci are now included in the Appendix.
- **Reviewer 2:** Following the suggestion to refine our fitness testing, we added a competition assay involving the wild-type *Comamonas testosteroni* CNB-2 and used this as a benchmark to normalize the fitness results of mutants, confirming the consistent trends as previously observed.

Detailed responses are shown in **blue font** and modified or supplementary sentences are in **red font**.

Reviewer #1:

Hui Lin et al investigate the evolution of antibiotic resistance in *Comamonas testosteroni* when exposed to sub-MIC concentrations of sulfamethoxazole (SMX).

The authors demonstrate that under these conditions, resistance evolves in two ways: a fast, reversible mechanism driven by DNA methylation of genes in central metabolic pathways, and a slower, stable mechanism involving mutations in metabolic genes. These changes lead to a metabolic shift from glycolysis to gluconeogenesis, which enhances bacterial growth and biomass production. The study also shows that increasing temperatures accelerate the evolution of resistance by promoting mutations in metabolic genes. The findings suggest that metabolic adaptations are relevant for antibiotic resistance, but it is difficult to distinguish cause and consequence from the data. Moreover, the role of global growth rate effects on both metabolic changes and antibiotic resistance is unclear. Thus, although the authors show interesting data and the role of metabolism for antibiotic action is a relevant topic, some points should be clarified to support the claims:

Main points:

- 1) The authors use the zero effective concentration point ZEP as a measure of resistance, but it remains unclear how ZEP directly relates to the MIC and why ZEP is a more informative measure. The authors should clarify the rationale behind using ZEP over MIC and provide data to support the claim that ZEP has better insights into resistance mechanisms. Otherwise replace ZEP with MIC, which is a common

measure for resistance also in the clinic. Also the authors should include how SMX levels relate to EUCAST breakpoints to understand clinical relevance.

Response: Thank you for your valuable feedback. We acknowledge the importance of the Minimum Inhibitory Concentration (MIC) as the standard clinical measure of antibiotic resistance. However, our study expands this framework by exploring the hormetic effects of antibiotics, characterized by a biphasic dose-response relationship that reverses bacterial growth response at low versus high concentrations. Hormesis studies commonly utilize the Zero Equivalent Point (ZEP), comparable to the No Observable Adverse Effect Level (NOAEL), to establish thresholds for safe doses and to assess the safety and acceptability of chemical exposures (Agathokleous, Saitanis et al., 2021, Calabrese, 1996, Kendig, Le et al., 2010). ZEP is defined as the highest dose at which a dose-response curve intersects the control-response line, as depicted in Figure EV1a. This intersection marks the threshold between beneficial and non-detrimental impacts, indicating the maximum dose of an antibiotic that bacteria can tolerate without significant adverse effects. Specifically, ZEP identifies a lower boundary where sub-MIC concentrations—our main focus and discussed area—begin to impact bacterial physiology, including metabolic activity, gene expression, and stress response mechanisms, without inhibiting growth. Thus, within the context of our study, ZEP emerges as a more fitting indicator of resistance, offering insights that extend our understanding beyond traditional metrics, which focus solely on complete growth inhibition.

As recommended, we have incorporated data on the concentration-response curves (CRCs) of evolved isolates under both static and dynamic conditions, as well as documenting changes in the MIC throughout evolution. Notably, the patterns observed in the MIC changes paralleled those seen for the ZEP. These results have been added to Expanded View Figures EV1 and EV2. Further details and discussions of these findings have been elaborated in the Results section as “To analyze resistance under the hormetic effects of antibiotics, we adopted the ZEP as a novel marker for bacterial resistance (Figure EV1a). This metric builds upon the No Observable Adverse Effect Level (NOAEL), marking the transition from stimulatory to neutral impacts of antibiotics on bacterial growth (Dorato & Engelhardt, 2005). The ZEP specifically identifies a threshold where SMX begins to modulate bacterial physiology—impacting metabolic activity, gene expression, and stress response mechanisms—without suppressing growth (Agathokleous et al., 2021, Calabrese, 1996, Kendig et al., 2010).” (page 6 from line 119 to line 126) and “Additionally, our analysis of changes in the MIC during the evolutionary process revealed a pattern of increase similar to that observed with the ZEP, further substantiating the progression of resistance (Figure EV2).” (page 7 from line 141 to line 143)

Figure EV1. Concentration-response curves (CRCs) for *Comamonas testosteroni* exposed to sulfamethoxazole (SMX). (a) Schematic representation of a CRC. Fitted CRCs of *C. testosteroni* during (b) static evolution and (c) dynamic evolution at different time points: days 15 (G150), 25 (G250), 35 (G350), and 55 (G550). Growth data for the evolved strains were normalized relative to the ancestral strain in an SMX-free environment, utilizing data from six biological replicates.

Figure EV2. Minimum Inhibitory Concentration (MIC) patterns observed over 550 generations of *Comamonas testosteroni* evolution under (a) static and (b) dynamic evolution protocols. Each evolutionary scheme comprised six biological replicates.

Moreover, our review of EUCAST breakpoint tables revealed no documented antibiotic susceptibility data for *C. testosteroni*. We have thoroughly discussed the implications of this absence, highlighting its clinical challenges. “This urgency is underscored by the clinical challenges posed by infections from *C. testosteroni*. Known for its prevalence in diverse environments, the broad adaptability of *C. testosteroni* makes it a potential agent for mild yet persistent infection in clinical settings (Farooq, Farooq et al., 2017, Farshad, Norouzi et al., 2012). This pathogen has been implicated in a wide array of conditions, from cellulitis to more severe infections such as peritonitis, endocarditis, and bacteremia (Orsini, Tam et al., 2014).

A major obstacle in treatment is the lack of established antibiotic susceptibility breakpoints for *C. testosteroni* in EUCAST guidelines. This absence necessitates reliance on broader-spectrum or empirical antibiotic strategies. These may not be optimal for treating *C. testosteroni*, which responds uniquely to the pressure of antibiotics. Additionally, the gradual decay of antibiotics in patients post-treatment creates an inconsistent concentration gradient, further complicating the effective management of these infections. Advancing research to uncover novel resistance mechanisms could lead to more targeted and effective treatment strategies, which are crucial for controlling the spread and evolution of antibiotic resistance in such atypical pathogens.” (page 26 from line 554 to line 567)

2) The metabolic changes and also the stimulatory effect, could potentially be caused by reduced bacterial growth rates under sub-MIC conditions rather than the direct effects of the antibiotics or the mutations. Antibiotic-based reduction of growth rates are known to strongly influence metabolism and proteome allocation (e.g. work of Terry Hwa DOI: 10.1126/science.1192588). The authors should describe how their gene expression and metabolism changes relate to such global growth effects. Further reduced growth rates are known to affect antibiotic susceptibility, and this could confound the conclusions. The authors need to report and analyze growth rates across all experimental conditions to rule out such general growth rate effects.

Response: Thank you for your insightful feedback. We have thoroughly measured and reported the growth rates of both ancestral strains and evolved isolates (G150, G250, G350, G550) under various evolution protocols across a range of SMX concentrations

from 0 to 100 mg·L⁻¹. The detailed methods are provided in the “*Measurement of Growth Rate*” section of the Materials and Methods. Our analysis showed that below the ZEP, both ancestral strains and evolved isolates exhibited higher per capita growth rates compared to SMX-free conditions. Moreover, isolates evolved under SMX consistently outperformed those evolved without SMX across all concentrations (Appendix Figure S2 and Appendix Table S1), with a significant positive correlation between growth rates and final biomass observed (Figure EV3a). These results eliminate the possibility that reduced growth rates could affect antibiotic susceptibility in our study, emphasizing the inherent evolutionary advantage of strains evolved under SMX exposure.

Importantly, the gene expression and metabolic changes observed in Figures 2-3, which compare SMX-evolved isolates to control evolved strains, were analyzed in an SMX-free environment. This confirms that these changes are inherent characteristics of the evolved strains, resulting from modifications and mutations in metabolic genes during evolution under SMX exposure. Recognizing the close association of global growth rate effects with metabolic changes and antibiotic resistance, our study highlights that metabolic gene mutations, acquired through evolutionary processes in antibiotic environments, directly contribute to increased resistance. These mutations drive a metabolic shift from glycolysis to gluconeogenesis, characterized by faster growth rates and higher biomass production.

We have included these findings in the Results section: “*We also measured per capita growth rates of both ancestral strains and evolved isolates across SMX*

concentrations from 0 to 100 mg·L⁻¹ to rule out the impact of reduced growth rates on SMX susceptibility. Results demonstrated that these isolates consistently exhibited higher growth rates below the ZEP compared to SMX-free conditions (Appendix Figure S2 and Appendix Table S1). Additionally, isolates evolved under SMX conditions consistently outperformed those evolved without SMX across all tested concentrations (Appendix Table S1). A significant positive correlation between per capita growth rate and biomass across all concentrations further confirmed that the observed enhanced resistance was not due to reduced growth rates (Figure EV3a), but rather reflected an inherent evolutionary advantage in strains evolved under SMX exposure. This advantage was further supported by our data, which demonstrated that biomass production and intrinsic growth rates of evolved isolates under both static and dynamic protocols consistently increased, surpassing those of control evolved strains despite stable nutrient conditions (Figures 1d-e, Appendix Figures S3-S4 and Table S1). A significant correlation between per capita growth rate and end biomass across all evolved isolates (G0-G550) was also observed (Figure EV3b). In contrast, control populations evolved in SMX-free conditions did not show increases in resistance levels or growth rates, suggesting that standard growth conditions, such as liquid medium and 24 h passaging, do not naturally facilitate the evolution of antibiotic resistance.” (page 8 from line 144 to line 161)

Appendix Table S1. Per capita growth rate (hr^{-1}) of *Comamonas testosteroni* cells to varying concentrations of sulfamethoxazole (SMX) during evolution on days 15 (G150), 25 (G250), 35 (G350), and 55 (G550). Red represents the ZEP range under the evolutionary scenario.

SMX concentration ($\text{mg}\cdot\text{L}^{-1}$)	0	0.005	0.0125	0.025	0.05	0.125	0.25	0.5	1	6.25	12.5	25	50	100	
Ancestral strain	0.51	0.53	0.55	0.58	0.61	0.57	0.55	0.48	0.43	0.35	0.18	0.12	0	0	
Control: Evolution with SMX-free	G150	0.55	0.58	0.60	0.61	0.64	0.65	0.6	0.51	0.45	0.43	0.21	0.15	0.05	0.04
	G250	0.51	0.53	0.55	0.57	0.63	0.53	0.52	0.51	0.41	0.4	0.31	0.18	0.09	0.01
	G350	0.53	0.55	0.58	0.59	0.64	0.60	0.53	0.49	0.42	0.35	0.20	0.11	0	0
	G550	0.56	0.59	0.61	0.62	0.68	0.64	0.63	0.54	0.49	0.38	0.19	0.14	0.08	0
SMX concentration ($\text{mg}\cdot\text{L}^{-1}$)	0	0.005	0.0125	0.025	0.05	0.125	0.25	0.5	1	6.25	12.5	25	50	100	
Static Evolution	G150	0.64	0.65	0.66	0.7	0.72	0.74	0.72	0.72	0.68	0.64	0.56	0.30	0.21	0.10
	G250	0.64	0.66	0.66	0.71	0.73	0.75	0.70	0.69	0.67	0.65	0.55	0.34	0.25	0.14
	G350	0.70	0.72	0.73	0.75	0.75	0.76	0.75	0.75	0.77	0.75	0.71	0.32	0.32	0.16
	G550	0.80	0.80	0.81	0.83	0.85	0.85	0.87	0.85	0.83	0.82	0.81	0.84	0.49	0.29
SMX concentration ($\text{mg}\cdot\text{L}^{-1}$)	0	0.005	0.0125	0.025	0.05	0.125	0.25	0.5	1	6.25	12.5	25	50	100	
Dynamic Evolution	G150	0.65	0.66	0.67	0.66	0.68	0.73	0.75	0.77	0.75	0.71	0.69	0.65	0.28	0.17
	G250	0.63	0.64	0.66	0.674	0.67	0.74	0.75	0.78	0.76	0.73	0.70	0.68	0.36	0.25
	G350	0.76	0.78	0.79	0.80	0.81	0.80	0.82	0.80	0.79	0.78	0.77	0.76	0.46	0.29
	G550	0.81	0.82	0.83	0.84	0.85	0.86	0.87	0.88	0.86	0.85	0.85	0.82	0.81	0.57
SMX concentration ($\text{mg}\cdot\text{L}^{-1}$)	0	0.005	0.0125	0.025	0.05	0.125	0.25	0.5	1	6.25	12.5	25	50	100	
Metabolic Evolution	G150	0.67	0.72	0.74	0.73	0.75	0.81	0.83	0.85	0.82	0.78	0.76	0.72	0.31	0.19
	G250	0.69	0.70	0.73	0.74	0.74	0.81	0.82	0.86	0.85	0.80	0.77	0.75	0.60	0.28
	G350	0.74	0.86	0.87	0.88	0.89	0.88	0.80	0.88	0.87	0.76	0.85	0.84	0.71	0.32
	G550	0.89	0.90	0.91	0.92	0.94	0.95	0.96	0.97	0.95	0.94	0.96	0.90	0.89	0.63

Appendix Figure S2. Growth curve of wild-type *Comamonas testosteroni* cultured in media containing sulfamethoxazole (SMX) concentrations below the Zero Effective Concentration Point (ZEP) of $250 \mu\text{g}\cdot\text{L}^{-1}$. Time series for the optical density (OD₆₀₀) of each species in monoculture is shown. Each evolutionary scheme comprised six biological replicates.

Figure EV3. Correlation between growth rate and biomass (OD₆₀₀) in evolved isolates. (a) Across a range of sulfamethoxazole (SMX) concentrations (0.005 to 100 mg · L⁻¹) in the evolved isolates G150, G250, G350, and G550; (b) Across all evolved isolates (G0 – G550, control, static, dynamic, and metabolic evolution) in SMX-free environments. Each panel included the regression line, Pearson correlation coefficient (R), and the *p*-value from the T-test, indicating the statistical significance of the correlations.

3) Related to point 3, the authors claim that better growth comes from more efficient resource allocation (e.g., line 260). However, as I understand they measure growth based on end ODs after 24 h. But this can be achieved by reduced growth rates that would lead to higher end ODs because it reduces overflow metabolism. The authors need quantitative data about physiologies, at least biomass specific growth rates to make such claims about efficiency of metabolism.

Response: Thank you for your insightful comments. In our methodology, the growth rate was quantified by setting $OD_{600} = 0.5$ as the exponential growth threshold based on the growth of the wild type *C. testosteroni*. We recorded the time taken to reach this threshold for each replicate ($n = 6$), providing a measure that reflects the per capita growth rate accurately. It was essential to consider the biological reality that numerous biological processes, especially cell growth and response to antibiotics, often exhibit an initial lag phase before evident changes take place (Fridman, Goldberg et al., 2014, Li, Qiu et al., 2016). This period reflects the time a biological system needs to adjust and initiate a response. For this reason, we preferred to use the time-to-threshold method, as it allowed us to represent the actual biological response more accurately in our experimental system and consider both the rate of growth and final biomass, providing a comprehensive view of growth dynamics. We have supplied a new subsection titled “*Measurement of growth rates*” in the Materials and Methods section as “Growth rate assessments for the evolved isolates were conducted every day under various evolutionary protocols, including control, static, dynamic,

and metabolic evolution. Cell cultures were initiated at an OD_{600} of 0.1 and incubated in 96-well NB plates under SMX-free conditions. The plates were shaken at 170 rpm and maintained at a constant temperature of 30°C. The plates were tightly sealed, and OD was monitored at a wavelength of 600 nm. Growth rates were measured by assuming exponential growth to a threshold of OD 0.5 and the effective growth rate was determined by calculating the ratio $\lg(OD_{\text{threshold}} / OD_i) / T_{\text{threshold}}$, as previously described (Lin, Wang et al., 2024). This time-to-threshold measurement accounts for lag times, as strains with a time lag will reach the threshold OD later than those without, despite having similar exponential growth rates. Additionally, this method was used to assess the growth rates of evolved isolates at specific time points—days 15, 25, 35, and 55—across a range of SMX concentrations from 0.005 to 100 $\text{mg}\cdot\text{L}^{-1}$.”

(page 29 from line 610 to line 622)

Based on this approach, Figures 1d-e showed the calculated intrinsic growth rates per capita of evolved strains under various evolutionary protocols, demonstrating a gradual increase during static and dynamic processes that consistently surpassed those of the control evolved strains (Appendix Figure S4 and Appendix Table S1). Moreover, growth rates of these evolved isolates, especially those evolved under SMX conditions, showed significant positive correlations with end biomass (Figure EV3b). This increase is directly attributable to mutations in the metabolic genes of SMX-evolved strains, which led to a metabolic shift from glycolytic pathways to the gluconeogenic pathway, enabling rapid growth rate and high biomass production.

Sub-MIC levels of antibiotics are known to induce complex responses in bacteria, balancing stimulatory and inhibitory effects depending on the degree of target inhibition (Sun, Calabrese et al., 2020). Consequently, when these SMX-evolved mutants were subsequently re-exposed to SMX with biphasic dose effects, their gluconeogenic metabolism allowed them to more effectively counteract the inhibitory effects compared to their ancestral strains. This adaptive advantage is further evidenced by the evolved isolates achieving a higher ZEP, indicating a broader range of antibiotic concentrations over which the bacteria can grow without experiencing lethal effects. We have supplied more explanations in the Results section as “...This advantage was further supported by our data, which demonstrated that biomass production and intrinsic growth rates of evolved isolates under both static and dynamic protocols consistently increased, surpassing those of control evolved strains despite stable nutrient conditions (Figures 1d-e, Appendix Figures S3-S4 and Table S1). A significant correlation between per capita growth rate and end biomass across all evolved isolates (G0-G550) was also observed (Figure EV3b). In contrast, control populations evolved in SMX-free conditions did not show increases in resistance levels or growth rates, suggesting that standard growth conditions, such as liquid medium and 24 h passaging, do not naturally facilitate the evolution of antibiotic resistance.” (page 8 from line 153 to line 161)

Appendix Figure S4. Growth curve was measured to determine the per capita growth rate of (a) G150, (b) G250, (c) G350 and (d) G550 evolved isolates in sulfamethoxazole (SMX)-free conditions. Time series for the optical density (OD_{600}) of each condition is shown, comprising six biological replicates.

Figure EV3. Correlation between growth rate and biomass (OD₆₀₀) in evolved isolates. (a) Across a range of sulfamethoxazole (SMX) concentrations (0.005 to 100 mg · L⁻¹) in the evolved isolates G150, G250, G350, and G550; (b) Across all evolved isolates (G0 – G550, control, static, dynamic, and metabolic evolution) in SMX-free environments. Each panel includes the regression line, Pearson correlation coefficient (R), and the *p*-value from the T-test, indicating the statistical significance of the correlations.

We acknowledge the potential confounding effects of overflow metabolism in our analyses. Indeed, *C. testosteroni* is distinguished by its nonfermentative chemoorganotrophic metabolism, and it has shown inactivity in oxidation-fermentation tests using glucose as a substrate (Kumar, Singh et al., 2023, Willems & De Vos, 2006). As a result, this strain lacks the common fermentation pathways that produce lactic acid, ethanol, or acetic acid, effectively eliminating these typical sources of overflow metabolism. To support this notion, we assessed additional physiological metrics in evolved strains, including oxygen consumption rate (OCR), NAD^+/NADH ratios, and key metabolic byproducts associated with overflow metabolism like acetate, ethanol, and lactate. These measurements clearly confirmed that the higher final OD values are not attributable to fermentation processes. We have included these additional findings in the Results section as “To address potential confounding factors of overflow metabolism, we quantified typical byproducts such as acetate, ethanol, and lactate in the G550 isolates. These compounds were undetectable in both static and dynamic conditions, suggesting no overflow metabolism, and all isolates showed increased growth rates and biomass yield (Appendix Table S4). Physiological assessments further confirmed that oxygen consumption rates did not decrease in these isolates (Appendix Figure S8a). Additionally, unlike the control evolved isolates, neither static nor dynamic isolates exhibited elevated NAD^+/NADH ratios, which typically indicate hyperactive TCA cycle activity under oxidative stress (Appendix Figure S8b). Taken together,

resistance-evolved cells undergo a critical metabolic shift from glycolysis to gluconeogenesis. This remodeling, along with the increased biosynthetic demands of gluconeogenesis, consumes both carbon and ATP, reducing the carbon available for metabolite secretion or futile cycling through the EMP and PP pathways (Figure 3e and Appendix Table S4). Consequently, mutations in metabolic genes allowed the resistance-evolved isolates to optimize their metabolic efficiency, thereby enhancing their intrinsic resistance.” (page 14 from line 286 to line 299)

Appendix Figure S8. Physiological characteristics of G550 evolved isolates. (a) oxygen consumption rate (OCR) and (b) [NAD⁺]/[NADH] redox ratio. Each measurement was biologically replicated three times.

Appendix Table S4 Physiological characteristics of 550 generations of *Comamonas testosteroni* evolution under static, dynamic, and metabolic evolution protocols. Data are expressed as mean \pm SD of three biological replicates.

Parameter	G550 evolved isolates		
	Control	Static	Dynamic
Growth rate (h ⁻¹): Rate of increase in biomass	0.56 \pm 0.01	0.8 \pm 0.02	0.81 \pm 0.01
Biomass yield: (g _{CDW} /g _{Substrate})	0.16 \pm 0.04	0.52 \pm 0.06	0.55 \pm 0.07
Metabolic secretions (mmol/g _{CDW} /h)	Product secretion rate		
α -Ketoglutarate	2.87 \pm 0.3	1.59 \pm 0.05	1.26 \pm 0.12

(α -KG)			
Citrate	1.76 \pm 0.04	ND	ND
Fumarate	ND	ND	ND
Malate	ND	ND	ND
Pyruvate	5.40 \pm 0.2 $\times 10^{-3}$	ND	ND
OAA	3.74 \pm 0.1 \times 10^{-2}	ND	ND
Glutamate	7.4 \pm 2.3 \times 10^{-3}	ND	ND
Acetate	ND	ND	ND
Ethanol	0.56 \pm 0.15	ND	ND
Lactate	ND	ND	ND

We have also supplied a new subsection titled “*Measurement of the Oxygen consumption rate (OCR)*” in the Appendix Method S3 as “OCR was quantified using an XFe96 Extracellular Flux Analyzer (Seahorse Bioscience). Overnight cultures of evolved isolates (G550) were first diluted in fresh NB media under SMX-free conditions until reaching an OD₆₀₀ of approximately 0.1. These cells were then further diluted to an OD₆₀₀ = 0.001, and 100 μ L of the diluted cells were seeded onto XF cell culture microplates that had been precoated with 15 μ L of poly-D-lysine (PDL) (Sigma). Following the addition of 100 μ L of fresh media to each well, cellular respiration was quantified. OCR measurements were conducted at 6-minute intervals, alternating between 3 minutes of measurement and 3 minutes of mixing. NAD⁺/NADH levels and byproducts of overflow metabolism were measured using liquid chromatography-tandem mass spectrometry as described previously.”

4) The authors claim that metabolic changes are a driving force behind the emergence of resistance. For example in the abstract “new genetic mechanisms of resistance

linked to metabolic remodeling”. However, the metabolic changes may also just be the consequence rather than the cause of resistance. The authors should provide additional evidence to support the claim that the metabolic changes are causal for resistance and not merely downstream effects of other resistance mechanisms. The most direct evidence would be to insert one of the metabolic mutations individually, although I understand that one mutation alone could be insufficient. Maybe another approach to show causality is to revert mutations in an evolved isolate to illustrate that they contribute directly to resistance.

Response: Thank you for your feedback. We have conducted additional experiments, reverting specific metabolic mutations in metabolically evolved isolates to their ancestral states using site-specific mutagenesis. Wild-type versions of mutated genes, carried on suicide plasmid pCVD442GS, were introduced into *E. coli* β 2155 donor strains, followed by conjugation with the recipient metabolically evolved isolates. By restoring the wild-type genes, we assessed the impact of original mutations on resistance by comparing antibiotic susceptibility and growth rates between the reverted strains and their mutated counterparts. The observed significant reduction in resistance upon reversing mutations in key metabolic genes confirms their direct contribution to resistance.

We have incorporated these findings into the Results section as follows: “To determine whether metabolic mutations directly confer resistance or are secondary effects of other mechanisms, we reversed mutations in specific metabolic genes (*TctA*,

Pck, *NadB*, *GltA*, *ALDH*, and *AcnB*) found in both classically evolved and metabolically evolved isolates. These mutations were reversed in metabolically evolved isolates, where SNPs in these genes had been identified. We employed the suicide plasmid pCVD442GS, which carries the wild-type versions of these genes, to revert the mutations in the evolved strains (Appendix Table S5). The plasmid facilitates the expression and integration of the wild-type genes into the genome of the recipient strain, *E. coli* β 2155 (Figure 4e), followed by conjugation with the metabolically evolved recipient strain. Subsequent comparisons of the ZEP between the reverted mutants and their metabolically evolved counterparts revealed significant reductions for all reverted mutants. Notably, there was a 16.98-fold decrease for the reverted mutants of the key cataplerotic enzyme in gluconeogenesis, *Pck*^M, and a 14.85-fold decrease for the reverted mutants of *AcnB*^M (Figure 4f). Per capita growth rate measurements further supported these results, with reverted mutants of *Pck*^M and *AcnB*^M displaying decreases of 1.52-fold and 1.49-fold, respectively (Figure 4g). These results confirmed that the identified metabolic mutations indeed directly contribute to resistance. Further investigations are required to determine whether these mutations modify catalytic activity or alter the structure and function of proteins.” (page 16 from line 333 to line 350)

Figure 4. Metabolic evolution directly enhances antibiotic resistance. (a) Schematic of metabolic evolution experiment: continuous sulfamethoxazole exposure with progressive metabolic activity increases, via daily temperature increments from 18 °C to 37 °C, over 55 days under conditions identical to those in Figure 1a. (b)-(c) Data depict a steady increase in zero effective concentration point (ZEP) and per capita growth rate in metabolically evolved isolates compared to ancestral strains (n = 6), outperforming standard static evolution. **The per capita growth rate referred to the population growth rate normalized by the initial population size.** (d) Mutation analysis showed increased frequencies of mutant alleles in core metabolic genes within populations from days 15 (G150), 35 (G350), and 55 (G550) (n = 3). **ZEP/ZEP₀** indicates the relative resistance levels of whole-genome tested isolates compared to control evolved isolates, determined across three replicates (n = 3). (e) Schematic of site-directed mutagenesis used to reverse mutations. Subfigures (f) and (g) displayed the ZEP and growth rates, respectively, of reversed metabolic gene mutants with

SNPs (*Pck*^M, *AcnB*^M, *TctA*^M, *NadB*^M, *ALDH*^M, and *GltA*^M). Significance levels were assessed using Wilcoxon test.

We also added a new subsection titled “*Mutation validation*” in the Appendix Method S4 as “Genes with known SNPs—*TctA*, *Pck*, *NadB*, *GltA*, *ALDH*, and *AcnB*—identified in metabolically evolved *C. testosteroni* mutants were targeted for reversion to their ancestral states. High-fidelity DNA polymerase was used to amplify the upstream and downstream sequences of each mutation site, which were fused via PCR to create homologous recombination arms. These fused arms were cloned into the suicide plasmid pCVD442GS, and the resulting plasmids were electroporated into *E. coli* β 2155 to create donor strains for each mutant gene. The donor strains were then used in conjugation experiments with metabolically evolved *C. testosteroni* mutants, and 50 μ L recombinants were selected on LB plates 30 μ g·mL⁻¹ containing gentamicin. These colonies represented single-crossover recombinant clones that integrated the complete suicide plasmid via homologous recombination at one arm, referred to as CNB-2/pCVD442GS-mutant. Following further culturing, double-crossover recombinant clones were selected, resulting from a secondary homologous recombination event that either introduced the desired mutation in the target gene or reverted the strain to its original form. To remove single-crossover clones, selected colonies from the gentamicin-resistant plates were streaked onto LB plates containing 10% sucrose and incubated overnight at 37°C/220 rpm. The clones were PCR-verified and sequenced to confirm the correct mutation. After three rounds

of streak plating, the final mutant strain was confirmed. Appendix Tables S7 and S12 list the strains, plasmids, and primers used in this study. All experiments were performed with $30 \mu\text{g}\cdot\text{mL}^{-1}$ gentamicin, and MICs were measured as previously described.”

Appendix Table S5 Reversed mutations (single nucleotide polymorphisms and InDels) of metabolically evolved populations.

Gene	Type	Descriptions	Mutation position	Nucleotide change	Codon change	Amino-acid change
TctA^M	SNP	substrate-binding protein	3510344	A- G	ATG(H)-GTG(H)	
			3510349	A- G	CTC(E)-CTT(K)	Glu-Lys
Pck^M	SNP	phosphoenolpyruvate carboxykinase	79893	G- T	GCC(A)-TCC(S)	Cys-Ala
			80072	T- G	CTT(L)-CTG(L)	
NadB^M	SNP	L-aspartate oxidase	4125170	G- A	GGC(G)-GAC(D)	Gly-Asp
GltA^M	SNP	citrate synthase	3888319	A- C	TTC(F)-TGC(C)	Phe-Thr
ALDH^M	SNP	aldehyde dehydrogenase	1275850	A- G , T- A	AAT(I)-GAA(F)	Phe-Ile
AcnB^M	SNP	aconitate hydratase	3800754	T- G	TTC(E)-GTC(D)	Glu-Asp

Appendix Table S7 Strains and plasmids used in this study.

Strain or plasmid	Strain	Relevant characteristics
Strain		
Comamonas testosteroni	CNB-2	Wild-type
	CNB-2/G150Mut	Evolved strain under control, static and dynamic protocols
	CNB-2/G250Mut	Evolved strain under control, static and dynamic protocols
	CNB-2/G350Mut	Evolved strain under control, static and dynamic protocols
	CNB-2/G550Mut	Evolved strain under control, static and dynamic protocols
	Metabolic variant	Isolated mutant carried mutations in TctA , Pck , NadB , GltA , Bug , Lpd , ALDH , and AcnB
	KF-1	Wild-type
Comamonas testosteroni reversed mutant	CNB-2 Metabolic variant - TctA ^M	CNB-2 metabolic mutants containing reversed genes TctA ^M
	CNB-2 Metabolic variant - Pck ^M	CNB-2 metabolic mutants containing reversed genes Pck ^M
	CNB-2 Metabolic variant - NadB ^M	CNB-2 metabolic mutants containing reversed genes NadB ^M
	CNB-2 Metabolic variant - GltA ^M	CNB-2 metabolic mutants containing reversed genes GltA ^M
	CNB-2 Metabolic variant - ALDH ^M	CNB-2 metabolic mutants containing reversed genes ALDH ^M
	CNB-2 Metabolic variant - AcnB ^M	CNB-2 metabolic mutants containing reversed genes AcnB ^M
Escherichia coli	β2155	Transconjugation donor: F' strA hsdS Δ(lacZ)M15 ΔdapA::erm pir::RP4(::kan from SM10)
Pseudomonas aeruginosa	PAO1	Wild-type
Plasmid		
pCVD442		Suicide vector, Gm ^R
pCVD442 - TctA ^M ::Gm		Vector pCVD442 containing Gm gene cassette-truncated reversed mutant gene with flanking sequences for generating mutant CNB-2 Metabolic variant - TctA ^M

pCVD442 - Pck^M::Gm	Vector pCVD442 containing Gm gene cassette-truncated reversed mutant gene with flanking sequences for generating mutant CNB-2 Metabolic variant - Pck^M
pCVD442 - NadB^M::Gm	Vector pCVD442 containing Gm gene cassette-truncated reversed mutant gene with flanking sequences for generating mutant CNB-2 Metabolic variant - NadB^M
pCVD442 - GltA^M::Gm	Vector pCVD442 containing Gm gene cassette-truncated reversed mutant gene with flanking sequences for generating mutant CNB-2 Metabolic variant - GltA^M
pCVD442 - ALDH^M::Gm	Vector pCVD442 containing Gm gene cassette-truncated reversed mutant gene with flanking sequences for generating mutant CNB-2 Metabolic variant - ALDH^M
pCVD442 - AcnB^M::Gm	Vector pCVD442 containing Gm gene cassette-truncated reversed mutant gene with flanking sequences for generating mutant CNB-2 Metabolic variant - AcnB^M

Appendix Table S12 Reversed mutations primers used in this study.

Primer	Description
Upstream homologous recombination arm primers (-5F, 5' Phosphorylated)	
TctA-5F	TTCACAGGAGGTGGAGTCTATG
TctA-5R	CACTCCACGCAGATCGGC
Pck-5F	CCTTCCTGGCCTGGACCG
Pck-5R	GAAATTGGTCTTGCCGCAGGC
NadB-5F	GATTTTGCCGCTGCCCAGC
NadB-5R	CGGCAGGCACAGCAAGGC
GltA-5F	GGCGACAAGGGTGAGCTG
GltA-5R	GGCTTCGTTGGCGCCGC
ALDH-5F	GGCATGTTTCTGTTCAACGGGCTGGAGAATCTGATGGG
ALDH-5R	GGCCCGGTCAAGGGCATCGTGCG
AcnB-5F	AGAGCAAAGTCGGCAAAGAACG
AcnB-5R	CGGTGCGGAGCGCAACAA
Downstream homologous recombination arm primers (-3R, 5' Phosphorylated)	
ALDH-3F	GCTCTCCTCGTGGAAGATGCGCATCGCC
ALDH-3R	GGCTTCGCTGACCACGCAGATCAATGGCG
TctA-3F	AGCAATGATGGGCTTGGTCCG
TctA-3R	TGACAGTCATGTTTCATGATGGTTTT
Pck-3F	ATGCTGGTGCCGCCCAAG
Pck-3R	CGTACTGGGGAGCAATGCC
NadB-5F	CTATGGGCAATTGGCGAGGT
NadB-5R	TTTTTGCTTTCCAGGGTGGCG
GltA-3F	GCGGCTTCCGTCGTGACG
GltA-3R	GGCCAGAGCGAAGATGCC
AcnB-3F	CGACAGGGGTTTGCTTCTTCG
AcnB-3R	GATGCGGTGTACTACCCCGA
Linker Primer (Red: Bases that have been substituted)	
TctA-linker	GTGCCGCGGATCTGCGTGGAGTG CTT AGCAATGATGGGCTTG GTCCGCT
ALDH-linker	CGCACGATGCCCTTGACCGGGCC AAG CTCTCCTCGTGGAAGATGCGCAT
Pck-linker	GGCCTGCGGCAAGACCAATTT TCC ATGCTGGTGCCGCCCAAGGCCTT
NadB-linker	GGAGCCTTGCTGTGCCTGCCG GAC CGGCACGGCCAGCCCCGGCGAGCG
GltA-linker	ACGGCGGGGCCAACGAAGCC TCG GCGGCTTCCGTCGTGACGCTCACCCC
AcnB-linker	CTCGGCCTTGTTGCGCTCCGCACCG GTC CGACAGGGGTTTGC TTCTTCG

5) Related to 4, did the authors find known SMX resistance mutations e.g. in the target gene or in transporters? To authors should at least better discuss the role of canonical resistance mechanism in their experiments.

Response: Thank you for your valuable feedback. We have discussed in the

Discussion section: “Despite the multifaceted roles of antibiotics, most resistance research has traditionally focused on their inhibitory effects, categorizing resistance mechanisms into three main categories: target modification, drug inactivation, and drug transport (Blair, Webber et al., 2015, Woodford & Ellington, 2007). Among these, target modification is the most commonly recognized mechanism of bacterial resistance to sulfonamides. According to the Comprehensive Antibiotic Resistance Database (CARD), mutations in the conserved regions of the dihydropteroate synthase gene (*folP*) are the predominant sulfonamide resistance mechanism (Sköld, 2000). These are closely followed by the highly mobile *sul* genes (*sulI-sul4*) that encode DHPS enzymes not inhibited by sulfonamides, identified in the genetic material of the surveyed 88 pathogens (Nunes, Manaia et al., 2020). These mutations, which have emerged across various bacterial lineages, are primarily chromosomal, although instances of horizontal gene transfer have been documented (Qvarnstrom & Swedberg, 2006). However, our extensive genetic analysis did not identify mutations in the genes that encode enzymes targeted by SMX, such as those within the *folP* and *sul* genes crucial for folate synthesis. By broadening our examination to include the potential stimulatory effects of antibiotics, our research proposes new resistance

mechanisms directly resulting from modifications and mutations in metabolic genes. This challenges traditional frameworks of resistance mechanisms, suggesting that existing categorizations may not fully capture all active mechanisms of action.” (page 21 from line 450 to line 469)

Minor:

In general the figure captions and method section should be better described to understand what data is shown. For example Figure 1d: please define per capita growth rate

Response: Thank you for your valuable feedback. The per capita growth rate refers to the population growth rate normalized by the initial population size. We have included a concise definition of the per capita growth rate in the Figure 1, 4, and 5 legend “The per capita growth rate referred to the population growth rate normalized by the initial population size.” Additionally, we have thoroughly revised the figure legend to improve the clarity and comprehensibility of the data presented.

According to Figure 1 and Figure S1 the authors measured MICs. Also, in Figure 3a it says MIC of n= 3 replicates but it is not clear which data this is. This is confusing and should be clarified.

Response: Thank you for pointing this out; we apologize for any confusion. We have updated the Figure 3 and 4 legends to more accurately convey this: “ZEP/ZEP₀ represents the relative resistance level of three whole-genome sequenced isolates compared to the control evolutionary strain.”

Reviewer #2:

Title: Epigenetic Modifications and Metabolic Gene Mutations Drive Resistance Evolution in Response to Stimulatory Antibiotics

Summary:

This study examined the evolution of antibiotic resistance under a stimulatory antibiotic. Experimental evolution of resistance to sulfamethoxazole in *C. testosteroni* CNB-2, whose growth is promoted by low doses of sulfamethoxazole (SMX), a fact found by a previous study by this group was performed. Populations taken at different stages of the evolution experiment were sequenced to trace SNPs that may contribute to SMX resistance. Once the evolution experiment was completed resistant isolates were subjected to a wide variety of analysis (MIC/ZEP determination, qRT-PCR, competition assays, growth assays, TEM, heat and oxidative stress assays, RNA-seq, and LC-MS/MS). All together this study suggests and supports novel mechanisms of resistance linked to metabolic remodeling for this system. There are two mechanisms a fast epigenetic response (though metalation) and a slower selection for mutations both of which occur in pathways related to metabolism. Both mechanism contribute to a transition from glycolysis to gluconeogenesis. The study also shows how an increase in environmental temperature can also accelerate the metabolic evolution by increasing the number of mutations in metabolic genes. However, these mutants are only favored in environments treated with SMX and are nearly or completely competitively excluded in both intra- and inter- specific co-cultures without SMX.

This work adds to bring light to the growing need for environmental factors, especially those related to climate change, to be taken into greater consideration when examining the antibiotic resistance crisis.

General remarks

Overall, the key conclusions are well supported. I hope that the trends from the competition assays remain once the WT CNB-2 experiments are performed (see the last major point for more details). This study is a perfect example of the need for more interdisciplinary approaches to the antibiotic resistance problem from an ecological and evolutionary standpoint. This unique study system allows for the authors to thoroughly investigate a previously overlooked aspect of a concentration response curve and shows novel antibiotic resistance mechanisms involving metabolism. The study will attract the interest of evolutionary, ecological, and clinical microbiologist along with those interested in public health.

Major points

- Lines: 68-80 - It would have been great to see how tolerance and persistence plays into these dynamics. Persistence is mentioned in the discussion, but it would be good to include it in the introduction

Response: Thank you for your valuable feedback. We have revised the introduction to incorporate a discussion on bacterial tolerance and persistence as “Various studies showed that the metabolic state of a bacterial cell influences its susceptibility to antibiotics, with certain states enhancing or reducing vulnerability (Gutierrez, Jain et al., 2017, Peng, Su et al., 2015, Zhao, Chen et al., 2021). Recent research provides

insights into the metabolic states of persisters within biofilms, which offer a protective niche against antibiotics and other adverse conditions (Davenport, Call et al., 2014, Donlan & Costerton, 2002). These cells often undergo metabolic downshifts, exacerbated by impaired nutrient penetration and consumption by peripheral cells, leading to reduced nutrient availability and possibly benefiting long-term survival through impaired metabolism. In *M. tuberculosis*, increased antibiotic tolerance and persistence have also been correlated with growth arrest and changes in carbon flux (Baek, Li et al., 2011).” (page 4 from line 70 to line 80)

• Lines: 109-110 - For the metabolic evolution experiment what is the rationale for the temperature range used? Were these temperatures the fundamental niche width of the thermal performance curve of the WT/ancestral strain?

Response: Thank you for your valuable feedback. The temperature range of 18°C to 37°C was derived from the growth rate response curve of wild-type *C. testosteroni* (Appendix Figure S9). This range supports enhanced metabolic activity, is crucial for identifying metabolic mutants, and reflects the organism's typical environmental conditions. We have clarified this in the Results section: “The optimal temperature range of 18°C to 37°C, based on the wild-type *C. testosteroni* growth rate response curve (Appendix Figure S9), aligns with the organism’s ecological tolerance and typical environmental conditions, enabling metabolic optimization to aid in identifying specific mutants.” (page 15 from line 313 to line 316)

Appendix Figure S9. The growth rate response across a temperature range of 17°C to 40°C. Each data point represents the mean of six biological replicate.

- Lines: 115-117 - It is unclear in the methods if the experimental evolution occurred with the MSM or succinate-MSM medium. After reading the results it was made clear when this medium was used but it would have been beneficial to have this more explicitly stated in the methods.

Response: Thank you for your valuable feedback. We have updated the Materials and Methods section to specify as “In addition to the NB medium utilized for evolutionary experiments, gluconate-mineral salt medium (MSM) and succinate-MSM were used to culture wild-type strains. This setup was intended to determine if wild-type strains could directly modify their metabolic strategies to enhance resistance by utilizing different carbon sources. Details of the culture media are presented in Appendix Table S9.” (page 28 from line 587 to line 592)

- Lines: 118-127 - Although references were provided from a previous study looking at the ZEP it would be great to have a brief summary of how the ZEP was determined and the definition of the ZEP.

Response: Thank you for your valuable feedback. We have provided additional explanations in the Materials and Methods section regarding the determination and definition of the ZEP as “The ZEP is defined as the highest antibiotic dose at which the CRC intersects the control-response line, indicating the maximum dose that *C. testosteroni* can tolerate without significant adverse effects (Dorato & Engelhardt, 2005). The ZEP was determined by analyzing CRCs, which show the growth response of *C. testosteroni* to varying concentrations of SMX. Data were fitted using a sigmoidal model employing the Hill inverse dichotomy iteration technique, enabling precise identification of the transition point. The model parameters, including the ZEP, were optimized using the Levenberg-Marquardt algorithm as described previously (Wang, Liu et al., 2018), enhancing the accuracy and reliability of our findings. Details are provided in the Appendix Method S1.” (page 28 from line 598 to line 607)

Moreover, we have elaborated in the Results section as “To analyze resistance under the hormetic effects of antibiotics, we adopted the ZEP as a novel marker for bacterial resistance (Figure EV1a). This metric builds upon the No Observable Adverse Effect Level (NOAEL), marking the transition from stimulatory to neutral impacts of antibiotics on bacterial growth (Dorato & Engelhardt, 2005). The ZEP specifically identifies a threshold where SMX begins to modulate bacterial physiology—impacting metabolic activity, gene expression, and stress response mechanisms—without suppressing growth (Agathokleous et al., 2021, Calabrese,

1996, Kendig et al., 2010). We determined the ZEP by analyzing CRCs using the Hill inverse dichotomy iteration model.” (page 6 from line 119 to line 126)

Figure EV1. Concentration-response curves (CRCs) for *Comamonas testosteroni* exposed to sulfamethoxazole (SMX). (a) Schematic representation of a CRC. Fitted CRCs of *C. testosteroni* during (b) static evolution and (c) dynamic evolution at different time points: days 15 (G150), 25 (G250), 35 (G350), and 55 (G550). Growth data for the evolved strains were normalized relative to the ancestral strain in an SMX-free environment, utilizing data from six biological replicates.

• Figure 1d - How were growth rates determined?

Response: Thank you for your insightful comments. The growth rate was quantified by designating $OD_{600} = 0.5$ as the exponential growth threshold based on the growth

of the wild type *C. testosteroni*. We then recorded the time taken to reach this threshold and calculated the average using all replicates (n = 6 for each member). Our experiment was designed to investigate how bacteria react to antibiotics. It was essential for us to consider the biological reality that numerous biological processes, especially cell growth and response to antibiotics, often exhibit an initial lag phase before evident changes take place (Fridman et al., 2014, Li et al., 2016). This period reflects the time a biological system needs to adjust and initiate a response. For this reason, we preferred to use the time-to-threshold method, as it allowed us to represent the actual biological response more accurately in our experimental system. We have added a new subsection titled “*Measurement of growth rates*” in the Materials and Methods section as “Growth rate assessments for the evolved isolates were conducted every day under various evolutionary protocols, including control, static, dynamic, and metabolic evolution. Cell cultures were initiated at an OD₆₀₀ of 0.1 and incubated in 96-well NB plates under SMX-free conditions. The plates were shaken at 170 rpm and maintained at a constant temperature of 30°C. The plates were tightly sealed, and OD was monitored at a wavelength of 600 nm. Growth rates were measured by assuming exponential growth to a threshold of OD 0.5 and the effective growth rate was determined by calculating the ratio $\lg(\text{OD}_{\text{threshold}} / \text{OD}_i) / T_{\text{threshold}}$, as previously described (Lin et al., 2024). This time-to-threshold measurement accounts for lag times, as strains with a time lag will reach the threshold OD later than those without, despite having similar exponential growth rates. Additionally, this method was used to assess the growth rates of evolved isolates at specific time points—days 15, 25, 35,

and 55—across a range of SMX concentrations from 0.005 to 100 mg·L⁻¹.” (page 29 from line 608 to line 620)

• Line: 472 - What was the rationale behind the species selection for interspecies competition assay? Did PAO1 have previous exposure to SMX? Did this strain of PAO1 recently evolve antibiotic resistance? Depending on this answer there might be compensatory mutations to maintain better growth thus making it a better competitor when your stain has not had this time yet. It would be good to know what the competition assays are between WT CNB-2 v KF-1 and WT CNB-2 v WT PAO1 to have a point of comparison. Then the results can be normalized to the WT CNB-2 results to see how the mutations may have affected the ability to compete in coculture.

Response: Thank you for your valuable feedback.

- **Species Selection:** We chose *Pseudomonas aeruginosa* PAO1 because it frequently coexists with *Comamonas testosteroni* in diverse environments such as activated sludge, marshes, and marine habitats (Chen, Fang et al., 2022, Li, Wang et al., 2022, Qing, Nicol et al., 2023, Wang, Ma et al., 2024). This frequent co-occurrence suggests potential natural interactions and competition. Moreover, PAO1 has robust resource acquisition abilities and an extensive arsenal of antimicrobial mechanisms, including bacteriocins and type VI secretion systems that inject toxins into competing cells (Ghequire & De Mot, 2014). These traits make PAO1 a formidable competitor, favoring aggressive over cooperative interactions. Our previous studies also demonstrated that PAO1 effectively engages in competitive interactions with *C. testosteroni*, particularly under the

influence of SMX, further supporting its selection for assessing interspecific fitness dynamics in our experiments (Lin et al., 2024).

- **Antibiotic Sensitivity of PAO1:** The *P. aeruginosa* PAO1 strain used in our assays was a glycerol-activated primary strain with no prior exposure to SMX, ensuring no pre-existing resistance mechanisms. We recorded the growth response of PAO1 at different SMX concentrations, all showing varying degrees of growth inhibition (Appendix Figure S14a). Moreover, the growth rate under $200 \mu\text{g}\cdot\text{L}^{-1}$ SMX used in interspecific fitness assays showed a slight decrease compared to the SMX-free environment (Appendix Figure S14b).

We have supplied more explanations in the Results section: “PAO1’s robust resource acquisition abilities, extensive antimicrobial arsenal, and frequent coexistence with *C. testosteroni* in diverse habitats make *P. aeruginosa* an ideal model for assessing interspecific fitness dynamics under competitive conditions (Ghequire & De Mot, 2014, Sana, Berni et al., 2016).” (page 19 from line 403 to line 407) and “To ensure that our findings were not influenced by pre-existing resistance mechanisms, we assessed the baseline susceptibility of PAO1 to SMX. With no prior exposure to the antibiotic, *P. aeruginosa* displayed typical growth inhibition across a range of SMX concentrations (Appendix Figure S14a). Notably, the growth rate decreased under a concentration of $200 \mu\text{g}\cdot\text{L}^{-1}$ SMX, demonstrating no significant resistance (Appendix Figure S14b). This supported the notion that the observed decrease in interspecific fitness among resistance mutants was not due to antibiotic

resistance in competing species but may be attributed to other, yet-to-be-identified, fitness costs.” (page 20 from line 412 to line 419)

Appendix Figure S14. Antibiotic susceptibility and growth dynamics of *Pseudomonas aeruginosa* PAO1. **(a)** Antibiotic susceptibility to sulfamethoxazole (SMX) expressed as optical density at 600 nm (OD₆₀₀), measured after 24 hours of growth in 96-well microtiter plates. Data represents mean ± standard deviation (SD) from six biological replicates, with the horizontal dashed line indicating the average cell density observed in the absence of SMX. **(b)** Growth curve showing the per capita growth rate of *P. aeruginosa* PAO1 at 200 µg·L⁻¹ SMX, used in interspecific fitness test.

- **Normalized fitness:** We have conducted additional competition assays between wild-type *C. testosteroni* CNB-2 and both KF-1 and *P. aeruginosa* PAO1. The results have been incorporated into an updated Figure 6a, which now shows normalized fitness values for the mutants. Normalized fitness was calculated using the formula:
$$\text{Normalized fitness} = \frac{\text{fraction mutant vs. competitor}}{\text{fraction WT CNB-2 vs. competitor}}$$
 This formula provides a quantitative measure of fitness, where values greater than 1 indicate superior fitness of the mutant strain relative to the wild type under competitive conditions, while values between 0 and 1 indicate reduced fitness. By applying standardized fitness metrics, we provide a clearer understanding of

how resistance mutations influence competitive interspecific interactions and confirm the consistent competitive outcomes observed previously.

The Results section has been revised to include: “To quantify competitive fitness,

the following formula was used: Normalized fitness =

$\frac{\text{fraction mutant vs. competitor}}{\text{fraction WT CNB-2 vs. competitor}}$ Values exceeding 1 (normalized fitness > 1) signify

enhanced fitness of the mutant strain relative to the wild type under competitive conditions, whereas values below 1 ($0 < \text{normalized fitness} < 1$) indicate reduced fitness. Under selective SMX pressure, mutants dominated both wild-type CNB-2 and KF-1 populations, constituting about 100% of the population after 24 hours, with a normalized fitness of 1.47 ± 0.22 (Figure 6a, Appendix Figure S12a and S13a-b).”

(page 19 from line 394 to line 401) and “...In competition experiments, the fraction of mutants significantly diminished to around 30%, with normalized fitness values of 0.62 ± 0.15 for G150, 0.43 ± 0.06 for G350, and 0.23 ± 0.08 for G550 (Figure 6a, Appendix Figure S12b and S13c, Wilcoxon $p < 0.01$). These findings suggested that while metabolic mutants maintain fitness within their own species, this advantage did not persist in mixed-species environments and may even lead to potential extinction.”

(page 20 from line 407 to line 412)

Appendix Figure S12. Competition assays between (a) *Comamonas testosteroni* CNB-2 wild-type and kin bacteria KF-1, or (b) *C. testosteroni* CNB-2 wild-type and out-group species *Pseudomonas aeruginosa* PAO1. Data represents mean \pm standard deviation (SD) from six biological replicates.

Figure 6. Normalized growth fitness of evolved resistant isolates and cross-resistance. (a) Growth competition between evolved resistant isolates, wild-type *Comamonas testosteroni* CNB-2, kin bacteria *C. testosteroni* KF-1, and out-group species *Pseudomonas aeruginosa* PAO1 in NB medium with and without antibiotics. Six parallel populations containing cells A and B were performed at a 1:1 ratio. The heatmap displayed normalized fitness values calculated using the formula:

$$\text{Normalized fitness} = \frac{\text{fraction mutant vs. competitor}}{\text{fraction WT CNB-2 vs. competitor}}$$

In this formula, “fraction”

referred to the ratio of cell density, specifically measured as colony-forming units per milliliter (CFUs/mL), of each cocultured strain. When the competitor is the wild-type CNB-2, the normalized fitness simplified to:

Normalized fitness = fraction mutant vs. WT CNB-2. (b) Evolved mutants were subjected to severe oxidative stress and heat shock to assess cross-resistance (n = 6). Significance levels were assessed using Wilcoxon test.

Minor points

- Line: 164 - At times the results section reads more like a discussion. This is purely a stylistic choice, but I would try and limit speculation/suggestions in the results section and keep it to the discussion section.

o Example: Lines 192-196

Response: Thank you for your valuable feedback. We have thoroughly revised the manuscript to refine the presentation of our results, ensuring that speculative and discussion-like expressions are appropriately confined to the discussion section. For instance, we clarified the observation in the results section: “The oscillatory resistance patterns observed in dynamically evolving strains demonstrated the emergence of acquired resistance, followed by its rapid reversal to susceptibility in new environments. This pattern suggested a potential transition from unstable, diverse epigenetic mechanisms to stable, mutation-induced intrinsic resistance.” (page 7 from line 137 to line 140)

- Line: 180 - typo. “resistant” should be “resistance”

Response: Revised.

- Figure 3d - It would have been nice if the color scheme was the same as Figure 3a & 3c

Response: Thank you for your valuable feedback. We have revised the color scheme in Figure 3d. We appreciate your attention to detail and believe this change improves the clarity and visual consistency of the figure.

- Line: 468 - What was the rationale for the strain selected in the competition assays? Are there any phenotypic differences between KF-1 and the WT CNB-2?

Response: Thank you for your valuable feedback. Research has shown that different genotypes of the same bacterial species (kin bacteria) often coexist closely and exhibit a broad spectrum of interactions, especially intense competition for identical nutrients and spaces (Keller & Surette, 2006, Koch, Germscheid et al., 2020, Palmer & Foster,

2022, Vaz Jauri & Kinkel, 2014). To investigate intra-specific competition fitness, we selected different genotypes of *C. testosteroni*, specifically KF-1, due to their 99% genetic identity and similar ecological niches, as confirmed by 16S rRNA gene analysis (Ma, Zhang et al., 2009). Both KF-1 and CNB-2 demonstrated similar growth characteristics and antibiotic susceptibilities (Appendix Figure S11), which simplified the analysis of intraspecific competition by minimizing confounding variables. Nevertheless, subtle differences in gene expression profiles related to stress responses may affect their competitive dynamics, potentially differentiating their ecological fitness. These significant genomic similarities, coupled with their phenotypic variations, provide a robust model for studying environmental adaptation and competitive behaviors.

We have supplied more explanation for the chosen of KF-1 strains as “The chosen KF-1 strain of *C. testosteroni* shares 99% genetic identity with CNB-2, confirmed by 16S rRNA gene analysis (Ma et al., 2009). Similar growth characteristics and antibiotic susceptibilities between KF-1 and CNB-2 enable accurate intra-specific competition dynamics analysis (Appendix Figure S11).” (page 19 from line 392 to line 394)

Appendix Figure S11. Growth curves for *Comamonas testosteroni* CNB-2 and KF-1 strains were measured under (a) SMX-free conditions and (b) 200 $\mu\text{g}\cdot\text{L}^{-1}$ SMX to determine their growth rates. The time series of optical density for each species in monoculture is presented with six biological replicates.

• Line: 531 - Are there any other known stimulatory effects observed between an antibiotic and bacterial species? It would be good to touch on this to help give additional relevance to the resistance mechanisms that are brought to light here.

Response: Thank you for your comment. We have expanded the Discussion to detail

how low doses of antibiotics stimulate various effects across multiple organisms as

“At low doses, antibiotics can induce a range of non-lethal, yet significant effects,

including extensive transcriptional reprogramming, stimulation of cell proliferation,

activation of silent natural product biosynthesis gene clusters, and modulation of

quorum sensing. These hormetic effects, which are critical stimulatory responses to

low-dose stressors such as antibiotics, oxygen, metals, or nanomaterials, play a crucial

role in the survival and adaptation of diverse biological entities, from microorganisms

to plants and animals (Calabrese, 2008, Davies, Spiegelman et al., 2006, Iavicoli,

Fontana et al., 2021).” (page 21 from line 444 to line 450)

• Lines: 586-609 - I would perhaps separate this out to two different paragraphs. I

would also suggest including the following citations:

o Rodríguez-Verdugo, A., Lozano-Huntelman, N., Cruz-Loya, M., Savage, V. and Yeh, P., 2020. Compounding effects of climate warming and antibiotic resistance. *IScience*, 23(4).

o Reverter, M., Sarter, S., Caruso, D., Avarre, J.C., Combe, M., Pepey, E., Pouyaud, L., Vega-Heredía, S., De Verdál, H. and Gozlan, R.E., 2020. Aquaculture at the crossroads of global warming and antimicrobial resistance. *Nature communications*, 11(1), p.1870.

o Rzymiski, P., Gwenz, W., Poniedziałek, B., Mangul, S. and Fal, A., 2024. Climate warming, environmental degradation and pollution as drivers of antibiotic resistance. *Environmental Pollution*, p.123649.

Response: Thank you for the suggestion to enhance clarity. We have revised the section into two distinct paragraphs and incorporated the recommended citation, to better support the discussion on temperature impacts on microbial resistance.

References

- Agathokleous E, Saitanis C, Markouizou A (2021) Hormesis Shifts the No-Observed-Adverse-Effect Level (NOAEL). *Dose-Response* 19: 15593258211001667
- Baek S-H, Li AH, Sasseti CM (2011) Metabolic regulation of mycobacterial growth and antibiotic sensitivity. *PLoS biology* 9: e1001065
- Blair JM, Webber MA, Baylay AJ, Ogbolu DO, Piddock LJ (2015) Molecular mechanisms of antibiotic resistance. *Nature Reviews Microbiology* 13: 42-51
- Calabrese EJ (1996) Expanding the RfD concept to incorporate and optimize beneficial effects while preventing toxic responses from nonessential toxicants. *Ecotoxicology and environmental safety* 34: 94-101
- Calabrese EJ (2008) Hormesis: Why it is important to toxicology and toxicologists. *Environmental Toxicology and Chemistry* 27: 1451-1474
- Chen C, Fang Y, Cui X, Zhou D (2022) Effects of trace PFOA on microbial community and metabolisms: Microbial selectivity, regulations and risks. *Water Research* 226: 119273
- Davenport EK, Call DR, Beyenal H (2014) Differential protection from tobramycin by extracellular polymeric substances from *Acinetobacter baumannii* and *Staphylococcus aureus* biofilms. *Antimicrobial agents and chemotherapy* 58: 4755-4761
- Davies J, Spiegelman GB, Yim G (2006) The world of subinhibitory antibiotic concentrations. *Current Opinion in Microbiology* 9: 445-453
- Donlan RM, Costerton JW (2002) Biofilms: survival mechanisms of clinically relevant microorganisms. *Clinical microbiology reviews* 15: 167-193
- Dorato MA, Engelhardt JA (2005) The no-observed-adverse-effect-level in drug safety evaluations: use, issues, and definition (s). *Regulatory toxicology and pharmacology* 42: 265-274
- Farooq S, Farooq R, Nahvi N (2017) *Comamonas testosteroni*: Is it still a rare human pathogen. *Case reports in gastroenterology* 11: 42-47
- Farshad S, Norouzi F, Aminshahidi M, Heidari B, Alborzi A (2012) Two cases of bacteremia due to an unusual pathogen, *Comamonas testosteroni* in Iran and a review literature. *The Journal of Infection in Developing Countries* 6: 521-525
- Fridman O, Goldberg A, Ronin I, Shoresh N, Balaban NQ (2014) Optimization of lag time underlies antibiotic tolerance in evolved bacterial populations. *Nature* 513: 418-421
- Ghequire MG, De Mot R (2014) Ribosomally encoded antibacterial proteins and peptides from *Pseudomonas*. *FEMS microbiology reviews* 38: 523-568

- Gutierrez A, Jain S, Bhargava P, Hamblin M, Lobritz MA, Collins JJ (2017) Understanding and sensitizing density-dependent persistence to quinolone antibiotics. *Molecular Cell* 68: 1147-1154. e3
- Iavicoli I, Fontana L, Agathokleous E, Santocono C, Russo F, Vetrani I, Fedele M, Calabrese EJ (2021) Hormetic dose responses induced by antibiotics in bacteria: A phantom menace to be thoroughly evaluated to address the environmental risk and tackle the antibiotic resistance phenomenon. *Science of the total environment* 798: 149255
- Keller L, Surette MG (2006) Communication in bacteria: an ecological and evolutionary perspective. *Nature Reviews Microbiology* 4: 249-258
- Kendig EL, Le HH, Belcher SM (2010) Defining hormesis: evaluation of a complex concentration response phenomenon. *International journal of toxicology* 29: 235-246
- Koch H, Germscheid N, Freese HM, Noriega-Ortega B, Luecking D, Berger M, Qiu G, Marzinelli EM, Campbell AH, Steinberg PD (2020) Genomic, metabolic and phenotypic variability shapes ecological differentiation and intraspecies interactions of *Alteromonas macleodii*. *Scientific Reports* 10: 809
- Kumar S, Singh O, Soni P, Juneja D, Yadav HK, Saleh OJ (2023) *Comamonas testosteroni*: A rare case of bacteremia in a patient with chronic liver disease.
- Li B, Qiu Y, Shi H, Yin H (2016) The importance of lag time extension in determining bacterial resistance to antibiotics. *Analyst* 141: 3059-3067
- Li Y, Wang F, Hou Z, Nie Z, Ma L, Hui S, Li D (2022) Microbiome in orbital fat under thyroid associated ophthalmopathy. *Medicine in Microecology* 13: 100058
- Lin H, Wang D, Wang Q, Mao J, Bai Y, Qu J (2024) Interspecific competition prevents the proliferation of social cheaters in an unstructured environment. *The ISME Journal* 18: wrad038
- Ma Y-F, Zhang Y, Zhang J-Y, Chen D-W, Zhu Y, Zheng H, Wang S-Y, Jiang C-Y, Zhao G-P, Liu S-J (2009) The complete genome of *Comamonas testosteroni* reveals its genetic adaptations to changing environments. *Applied and environmental microbiology* 75: 6812-6819
- Nunes OC, Manaia CM, Kolvenbach BA, Corvini PF-X (2020) Living with sulfonamides: a diverse range of mechanisms observed in bacteria. *Applied microbiology and biotechnology* 104: 10389-10408
- Orsini J, Tam E, Hauser N, Rajayer S (2014) Polymicrobial bacteremia involving *Comamonas testosteroni*. *Case reports in medicine* 2014
- Palmer JD, Foster KR (2022) Bacterial species rarely work together. *Science* 376: 581-582

- Peng B, Su Y-b, Li H, Han Y, Guo C, Tian Y-m, Peng X-x (2015) Exogenous alanine and/or glucose plus kanamycin kills antibiotic-resistant bacteria. *Cell metabolism* 21: 249-262
- Qing C, Nicol A, Li P, Planer-Friedrich B, Yuan C, Kou Z (2023) Different sulfide to arsenic ratios driving arsenic speciation and microbial community interactions in two alkaline hot springs. *Environmental Research* 218: 115033
- Qvarnstrom Y, Swedberg G (2006) Variations in gene organization and DNA uptake signal sequence in the *folP* region between commensal and pathogenic *Neisseria* species. *BMC microbiology* 6: 1-9
- Sana TG, Berni B, Bleves S (2016) The T6SSs of *Pseudomonas aeruginosa* strain PAO1 and their effectors: beyond bacterial-cell targeting. *Frontiers in cellular and infection microbiology* 6: 61
- Sköld O (2000) Sulfonamide resistance: mechanisms and trends. *Drug resistance updates* 3: 155-160
- Sun H, Calabrese EJ, Lin Z, Lian B, Zhang X (2020) Similarities between the Yin/Yang Doctrine and Hormesis in Toxicology and Pharmacology. *Trends in Pharmacological Sciences* 41: 544-556
- Vaz Jauri P, Kinkel LL (2014) Nutrient overlap, genetic relatedness and spatial origin influence interaction-mediated shifts in inhibitory phenotype among *Streptomyces* spp. *FEMS microbiology ecology* 90: 264-275
- Wang B, Ma B, Zhang Y, Stirling E, Yan Q, He Z, Liu Z, Yuan X, Zhang H (2024) Global diversity, coexistence and consequences of resistome in inland waters. *Water Research* 253: 121253
- Wang Z-J, Liu S-S, Qu R (2018) JSFit: a method for the fitting and prediction of J- and S-shaped concentration–response curves. *RSC advances* 8: 6572-6580
- Willems A, De Vos P (2006) *Comamonas*. In *The Prokaryotes: a handbook on the biology of bacteria*, pp 723-736. Springer
- Woodford N, Ellington MJ (2007) The emergence of antibiotic resistance by mutation. *Clinical Microbiology Infection* 13: 5-18
- Zhao X-l, Chen Z-g, Yang T-c, Jiang M, Wang J, Cheng Z-x, Yang M-j, Zhu J-x, Zhang T-t, Li H (2021) Glutamine promotes antibiotic uptake to kill multidrug-resistant uropathogenic bacteria. *Science Translational Medicine* 13: eabj0716

23rd Dec 2024

Manuscript Number: MSB-2024-12544R

Title: Epigenetic Modifications and Metabolic Gene Mutations Drive Resistance Evolution in Response to Stimulatory Antibiotics

Dear Dr. Bai,

Thank you for the submission of your revised manuscript to Molecular Systems Biology. We have now received the enclosed reports from the referees that were asked to re-assess it. As you will see the reviewers are now globally supportive and I am pleased to inform you that we will be able to accept your manuscript pending the following final amendments:

1) Please provide an updated email address for author Lutong Yang in our submission system, as the emails sent to the address provided have not been deliverable (ltyang_st@rcees.ac.cn).

2) Please format the Data availability section including the direct links to the accession codes according to the example below:

"The datasets and computer code produced in this study are available in the following databases:

- Chip-Seq data: Gene Expression Omnibus GSE46748 (<https://www.ncbi.nlm.nih.gov/geo/query/acc.cgi?acc=GSE46748>)

- Modeling computer scripts: GitHub (<https://github.com/SysBioChalmers/GECKO/releases/tag/v1.0>)

- [data type]: [full name of the resource] [accession number/identifier] ([doi or URL or identifiers.org/DATABASE:ACCESSION])"

3) Author contributions: Please remove it from the manuscript and specify author contributions in our submission system.

CRedit has replaced the traditional author contributions section because it offers a systematic machine-readable author contributions format that allows for more effective research assessment. You are encouraged to use the free text boxes beneath each contributing author's name to add specific details on the author's contribution. More information is available in our guide to authors:

<https://www.embopress.org/page/journal/17574684/authorguide#authorshippingguidelines>

4) Our journal encourages inclusion of *data citations in the reference list* to directly cite datasets that were re-used and obtained from public databases. Data citations in the article text are distinct from normal bibliographical citations and should directly link to the database records from which the data can be accessed. In the main text, data citations are formatted as follows: "Data ref: Smith et al, 2001" or "Data ref: NCBI Sequence Read Archive PRJNA342805, 2017". In the Reference list, data citations must be labeled with "[DATASET]". A data reference must provide the database name, accession number/identifiers and a resolvable link to the landing page from which the data can be accessed at the end of the reference. Further instructions are available at .

5) In the Methods, please take care of the following:

- The Materials and Methods section should be renamed to "Methods".

- Please ensure that a statement on whether or not blinding was done is included in the Methods even if no blinding was done. Please also be sure to update the Author Checklist with this information and where it can be found in the manuscript.

6) Please place individual sections of the manuscript in the following order: Title page - Abstract & Keywords - Introduction - Results - Discussion - Methods - Data Availability - Acknowledgements - Disclosure and Competing Interests Statement - References - Figure Legends - Expanded View Figure Legends.

7) For the figures and figure legends, please take care of the following:

- Please make sure to update the callouts of all figures in the main manuscript text. Currently callouts are missing for Appendix Table S12. The callout for Table S1 should be corrected to Appendix Table S1.

- Please remove the EV figures from the main manuscript. These should be uploaded as individual, high-resolution files. The legends should stay in the manuscript, with the heading Expanded View Figures Legends, and placed after the main figure legends. Please check "Author Guidelines" for more information:

<https://www.embopress.org/page/journal/17574684/authorguide#figureformat>

- Figure Legends (main + EV): 1. Please note that the exact p values are not provided in the legends of figures 3C, 4F, 6B.

- Please indicate the statistical test used for data analysis in the legends of figures 2A, 3C.

- Please note that information related to n is missing in the legends of figures 4F, G.

- Please note that the error bars are not defined in the legends of figures 1E, 3C, 4F, G; 5B, 6B.

- Please note that for heatmap present in figure 3A a numbered scale bar is not provided. This needs to be rectified.

- Please note that the yellow arrows are not defined in the legend of figure 2D, 5C. This needs to be rectified.

8) In a standard check for plagiarism, we noted that there is overlap between the following sentence and a previously published work, which we would suggest that you edit: "the relative abundance of the cataplerotic enzyme at the node between PEP and OAA"

9) Appendix file: In the Appendix file, please recolor the red font to black, as this file is not edited/typset.

10) Synopsis:

- Synopsis image: Please remove it from the synopsis text document and upload it as a separate high-resolution jpeg file. Please ensure that the dimensions are 550 pixels wide x (300-600) pixels high.

11) As part of the EMBO Publications transparent editorial process initiative (see our policy here:

https://www.embopress.org/transparent-process#Review_Process), Molecular Systems Biology will publish online a Peer

Review File (PRF) to accompany accepted manuscripts. This file will be published in conjunction with your paper and will include the anonymous referee reports, your point-by-point response and all pertinent correspondence relating to the manuscript. Let us know whether you agree with the publication of the PRF and as here, if you want to remove or not any figures from it prior to publication. Please note that the Authors checklist will be published at the end of the PRF.

12) Please provide a point-by-point letter INCLUDING my comments and your detailed responses (as Word file).

I look forward to reading a new revised version of your manuscript as soon as possible.

Yours sincerely,

Poonam Bheda, PhD
Scientific Editor
Molecular Systems Biology

Reviewer #1:

The authors have addressed all my point. I support the publication of the manuscript.

Reviewer #2:

By including added methods and the additional experiments, the authors have improved the manuscript and clarified the ZEP, which will particularly help readers from other fields appreciate the results. I have no further suggestions and congratulate the authors on a nice manuscript.

The authors addressed the remaining editorial issues.

7th Jan 2025

Manuscript number: MSB-2024-12544RR

Title: Epigenetic Modifications and Metabolic Gene Mutations Drive Resistance Evolution in Response to Stimulatory Antibiotics

Dear Dr. Bai,

Thank you again for sending us your revised manuscript. We are now satisfied with the modifications made and I am pleased to inform you that your paper has been accepted for publication.

Yours sincerely,

Poonam Bheda, PhD
Scientific Editor
Molecular Systems Biology
